# Programmed microalgae-gel promotes chronic wound healing in diabetes

Yong Kang ®[1], Lingling Xu[1], Jinrui Dong[1], Xue Yuan[1], Jiamin Ye[1], Yueyue Fan[1], Bing Liu[2] ✉, Julin Xie[3] ✉ & Xiaoyuan Ji ®[1,4] ✉

Chronic diabetic wounds are at lifelong risk of developing diabetic foot ulcers owing to severe hypoxia, excessive reactive oxygen species (ROS), a complex inflammatory microenvironment, and the potential for bacterial infection. Here we develop a programmed treatment strategy employing live Haematococcus (HEA). By modulating light intensity, HEA can be programmed to perform a variety of functions, such as antibacterial activity, oxygen supply, ROS scavenging, and immune regulation, suggesting its potential for use in programmed therapy. Under high light intensity (658 nm, 0.5 W/cm²), green HEA (GHEA) with efficient photothermal conversion mediate wound surface disinfection. By decreasing the light intensity (658 nm, 0.1 W/cm²), the photosynthetic system of GHEA can continuously produce oxygen, effectively resolving the problems of hypoxia and promoting vascular regeneration. Continuous light irradiation induces astaxanthin (AST) accumulation in HEA cells, resulting in a gradual transformation from a green to red hue (RHEA). RHEA effectively scavenges excess ROS, enhances the expression of intracellular antioxidant enzymes, and directs polarization to M2 macrophages by secreting AST vesicles via exosomes. The living HEA hydrogel can sterilize and enhance cell proliferation and migration and promote neoangiogenesis, which could improve infected diabetic wound healing in female mice.

Approximately 537 million people worldwide suffer from diabetes, and its incidence is projected to increase by 46% by 2045[1]. A quarter of diabetic patients have a lifetime threat of persistent nonhealing wounds such as diabetic foot ulcers, which force patients to be in danger of amputation, and 68% of these individuals have a life expectancy of less than 5 years[2]. Studies have shown that damaged neovascularization in response to hypoxia is one of the most severe causes of chronic wound deterioration in patients with diabetes[3–5]. Exposure to high glucose triggers rapid posttranslational hydroxylation and degradation of hypoxia-inducible factor-1alpha. When responding to soft tissue ischemia, this alteration renders diabetic wounds incapable of upregulating vascular endothelial growth factor

(VEGF), creating obstacles to angiogenesis and wound healing[6]. Additionally, excessive reactive oxygen species (ROS) are another important impediment to the diabetic wound healing process, causing perpetual and irreversible damage to biomolecules and sustaining macrophages in the M1 phenotype to exacerbate the inflammatory response[7–9]. As a result, the recruitment of M1 macrophages creates a unique injury microenvironment in wounds characterized by augmented proteolysis and oxidative cellular damage[9,10]. Moreover, open wounds are highly susceptible to bacterial infection, which exacerbates wound hypoxia and inflammation[11,12]. Thus, designing an all-rounder with space-time controllable oxygen release, ROS scavenging, modulation of

[1]Academy of Medical Engineering and Translational Medicine, Medical College, Tianjin University, Tianjin 300072, China. [2]Department of Disease Control and Prevention, Rocket Force Characteristic Medical Center, Beijing 10088, China. [3]Department of Burns, The First Affiliated Hospital of Sun Yat-Sen University, Guangzhou 510080, China. [4]Medical College, Linyi University, Linyi 276000, China. ✉e-mail: neaucn@126.com; xiejulin@mail.sysu.edu.cn; jixiaoyuan@tju.edu.cn

macrophage polarization, and antibacterial properties is a hot topic for the clinical treatment of chronic diabetic wounds.

Hydrogels are three-dimensional cross-linked polymer networks that can absorb and retain a large amount of water or biological fluids. They have a range of physical and chemical properties that make them suitable for various applications in different fields in real life[13,14]. Hydrogels have been extensively applied in the biomedical field due to their biocompatibility. They can be used as scaffolds for tissue engineering, drug delivery systems, and contact lenses. Hydrogels can also be engineered to respond to external stimuli such as temperature, pH, or light, enabling controlled drug release. In agriculture, hydrogels can be incorporated into soil to improve water retention and nutrient availability for plants. They help reduce water usage, prevent soil erosion, and promote plant growth by providing a favorable environment for roots[15]. In the environmental remediation field, hydrogels can be utilized for wastewater treatment and environmental cleanup. They can absorb and remove contaminants from water, including heavy metals and organic pollutants[16]. Hydrogels are considered promising wound dressings because of their softness comparable to that of the extracellular matrix (ECM) and their ability to mildly absorb exudates[17–21]. Due to their porous microstructure, hydrogels in particular have a high loading capacity[22–25]. Therefore, it is highly desirable to construct a hydrogel dressing loaded with multifunctional materials or drugs to simultaneously address the aforementioned issues[26,27]. In fact, a variety of multifunctional hydrogels for wound repair have been developed that contain high amounts of oxygen for resolving hypoxia, polyphenolic substances or nanozymes for ROS scavenging, and macrophage polarity-modulating molecules for regulating the immune microenvironment[20,27–33]. For example, Zhao et al. developed a therapeutic wound dressing, namely, MnCoO@PDA/CPH, utilizing a biomimetic hydrogel system and modified hydrogen peroxide-mimicking nanozymes. The hydrogel is engineered to simultaneously match the mechanical and electrical signals of the skin while possessing oxidative capability activated by $H_2O_2$[34]. Wu et al. prepared a versatile dynamic Schiff base and borate ester cross-linked glycopeptide hydrogel that could continuously generate oxygen, promote M2 polarization of macrophages, and eliminate reactive oxygen and nitrogen species[35]. Zhang et al. prepared an injectable hydrogel based on platelet-rich plasma and laponite that could accelerate wound healing by promoting macrophage polarization and angiogenesis in full-thickness skin[36]. Clearly, the current design for multifunctional hydrogels entails significant issues, such as complex separation, tedious preparation, low synergistic efficiency, and limited space-time control. Therefore, there is an urgent need for a hydrogel dressing with a simple composition but a procedural therapeutic strategy. Gelatin methacryloyl (GelMA) is a dual-functionalized gelatin obtained through the reaction between aminosylated gelatin and methacrylic anhydride. The abundant amino groups distributed along the main chain of gelatin provide rich reactive sites for methacrylic anhydride. Methacrylic anhydride, bound to amino groups, can further react with each other to form three-dimensional structures suitable for cell growth and differentiation in scientific research related to technology. GelMA has been demonstrated to possess excellent biocompatibility, cell adhesion properties, and mechanical performance. It is widely applied in tissue engineering, drug delivery, 3D printing, and other fields[37,38].

Comparing the recombination of active components to a single microbial active cell, the latter offers a wider range of functions and is more easily programmable[39–42]. Microorganisms have played a crucial role in advancing agricultural, industrial, and public health research by providing humans with advanced principles and methodologies[43–48]. First, in cases of nonsymbiosis, various types of microorganisms engage in interactions that inhibit the growth of one another, thereby contributing to antibacterial activity in wound healing. Second, oxygen plays a critical role in multiple aspects of wound healing, such as cell

proliferation, migration, adhesion, blood vessel development, and tissue regeneration. It is widely believed that microorganisms, such as Chlorophyta, Haematococcus (HEA), and Archaea, played a fundamental role in the emergence and evolution of life on Earth by producing oxygen through photosynthesis. Rather than relying on conventional oxygen supply materials, such as hemoglobin, metal oxides, peroxides, and other oxygen delivery systems[49–54], which have limited and uncontrollable load capacities and release rates, these organisms are photoautotrophic. They utilize photosystems for photosynthesis, enabling them to produce oxygen. For example, Zhou et al. developed a bioactive hydrogel strategy for inhibition of methicillin-resistant *Staphylococcus aureus*. The bioactive hydrogel was composed with the berberine loaded living microalga *Spirulina platensis* and carboxymethyl chitosan/sodium alginate, which could release BBR and $O_2$ and produce ROS, mediating chemophotodynamic therapy for methicillin-resistant *Staphylococcus aureus* infection[55]. In addition, numerous metabolites and cells from microorganisms can regulate the immune microenvironment and act as antioxidants; these substances are also used in a variety of food and drug additives[56]. Astaxanthin, a metabolite of HEA (3,3'-dihydroxy-, -carotene-4,4'-dione, AST), has high ROS scavenging and antioxidant activity[57–59]. It has been reported that ASTs, such as monothiol glutaredoxin, glutaredoxin (GRX), thioredoxin (TRX), thioredoxin reductase (TrxR), ferritin, monodehydroascorbate reductase (MDAR), peroxiredoxin (PrxR), glutathione peroxidase (GPX), glutathione reductase (GR), catalase (CAT), superoxide dismutase (SOD), and ascorbate peroxidase (APX), have various ROS scavenging enzyme activities[60,61]. In addition, AST can inhibit the expression of inflammatory cytokines (COX-2, TNF-α, IL-6, and IL-1β)[62]. Consequently, ASTs possess considerable potential and offer promising applications in human health against a broad spectrum of diseases. AST upregulates low-density lipoprotein receptor-related protein-1, which can bind to and downregulate p65, p-c-Jun, and NF-Kb, thereby blocking activation of the JNK and NF-κB pathways. Inhibiting the activation of JNK1 and NF-κB and promotes M2 polarization of macrophages[63]. In recent years, there has been an increasing interest in utilizing microalgae, such as HEA, for the production of the natural antioxidant AST. This is primarily attributed to their faster growth rates and higher biomass production capabilities[57,64–66]. Additionally, HEA is thought to possess the greatest ability to accumulate natural AST under conditions of environmental stress[67–70]. Based on the progressive and controllable oxygen evolution, as well as AST release capabilities of HEAs, there is tremendous long-term potential for their application in artificial scaffolds for complex tissue healing, myocardial oxygenators, and as adjuncts in tumor therapy. As far as we know, there have been no reports of utilizing microorganisms for progressive oxygenation and microenvironmental regulation in the treatment of chronic diabetic wounds.

Here, we develop a simple hydrogel by encapsulating active HEA cells in conventional GelMA gel and established a system for local procedural therapy (Fig. 1). To increase the efficiency of oxygen and AST release, HEA spheroplasts are first prepared, loaded with GelMA solution, and photocured in situ to better establish irregular wounds. According to the light power density, an abundance of chlorophyll mediates the transition between oxygen production via photosynthesis and heat production via the photothermal conversion effect in green HEA (GHEA) cells. Therefore, in the initial phase of treatment, efficient photothermal conversion under 658 nm laser irradiation with a 0.5 W/cm² power density can swiftly eliminate bacteria. At the second stage of treatment, severe hypoxia in the wound is resolved by providing a constant oxygen supply for photosynthesis under 658 nm laser irradiation with a 0.1 W/cm² power density. With deterioration of the culture environment (bound in hydrogel) or external stimulation (laser irradiation), AST gradually accumulates, turning HEA cells red (RHEA) and endowing them with a potent capacity to modulate the wound microenvironment. Subsequently, HEA cells routinely release

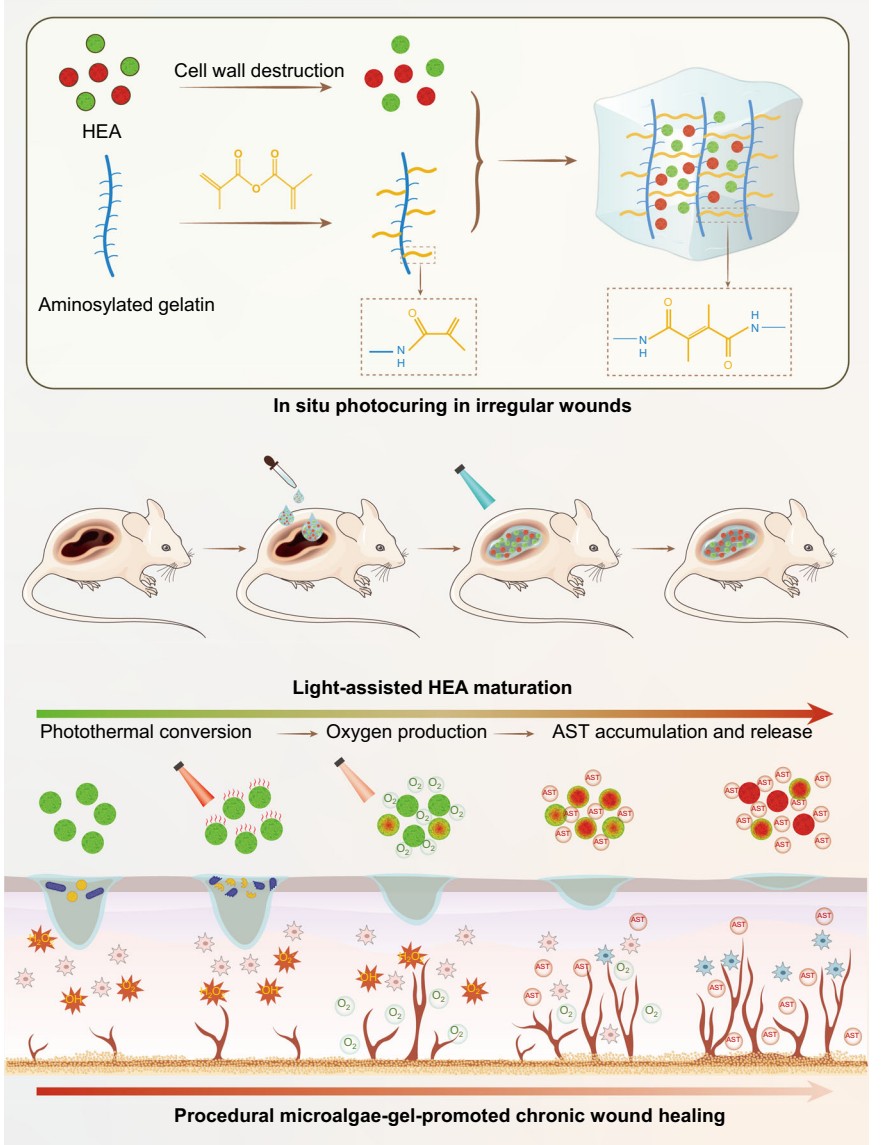

**Fig. 1 | Schematic illustration of the preparation and mechanism of the HEA@Gel-based wound healing strategy.** Irregular wound dressings were obtained by in-situ photocuring. By modulating light intensity, HEA@Gel can be programmed to perform antibacterial activity, oxygen supply, ROS scavenging, and immune regulation. HEA Haematococcus, AST astaxanthin.

AST to persistently scavenge ROS and regulate macrophage polarization. The healing of infected diabetic wounds is significantly accelerated by an all-in-one procedure that eliminates bacteria, supplies oxygen, scavenges ROS, promotes angiogenesis and facilitates anti-inflammatory effects via the polarization of M2 macrophages.

## Results

### HEA@Gel preparation and characterization

We designed a living microalgal-based procedural hydrogel dressing (HEA@Gel) to easily mask the wound and deliver soluble oxygen and AST into the wound bed via in situ photocuring (Fig. 2a). To verify the concept of procedural therapy, the most obvious evidence, the change in color of HEA cells, was identified and is displayed in Fig. 2b–e. When the necessary nutrients are plentiful, HEA cells grow rapidly and appear green due to the presence of chlorophyll (Fig. 2b). With the consumption of nutrients and exposure to light, the growth rate of HEA cells decreased, and AST gradually accumulated (Fig. 2c, d). As AST continued to accumulate, the HEA cells became distinctly darker red (Fig. 2e). The solid and thick cell wall of HEA cells results in their low permeability, which prevents AST from being effectively released

from within the cells (Fig. 2f)[71–73]. To remove the walls of HEA cells, we prepared HEA protoplasts using a mild enzymatic method (cellulase and pectinase). As shown in Fig. 2g and Supplementary Fig. 1, HEA-treated protoplasts retained their morphology, structure and activity after their cell walls were removed. Interestingly, after a period of culture, HEA protoplasts secreted numerous 200 nm diameter vesicles, which promoted the effective release and delivery of hydrophobic AST (Fig. 2h, i). To demonstrate the presence of AST within secretory vesicles, the vesicles were stained with DiI dye for membrane labeling. Due to the inherent fluorescence of AST, AST and the vesicles were observed under a laser confocal microscope, as shown in Supplementary Fig. 2, revealing clear colocalization of AST and the vesicles. To better compare the effects of the presence of the cell wall on AST release, equal amounts of HEA (with the cell wall) and HEA protoplasts (without the cell wall) were separately dispersed in 100 mL of deionized water. The released AST content was measured by using the standard curve of AST shown in Supplementary Fig. 3. As shown in Supplementary Fig. 4a, the rate of AST release from HEA protoplasts was significantly greater than that from HEA protoplasts with retained cell walls. Furthermore, over time, there was a continuous increase in

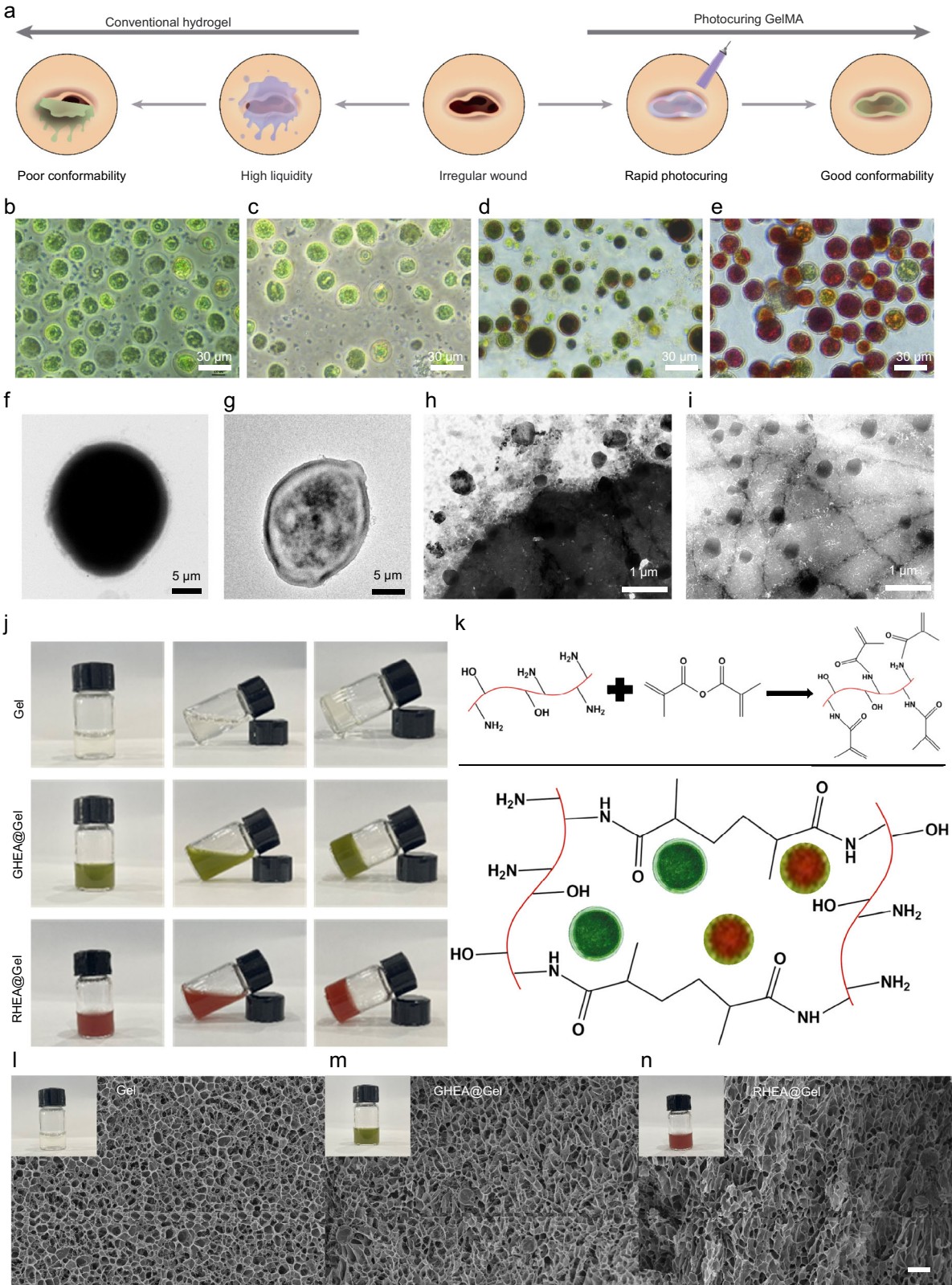

**Fig. 2 | HEA@Gel preparation and characterization. a** Schematic representation of the benefits of photocuring GelMA. **b**–**e** Photographs of various stages of the HEA. **f** TEM image of the HEA. **g** TEM image of HEA protoplasts. **h**–**i** TEM images of AST vesicles released from HEA. **j** Photographs of Gel, GHEA@Gel, and RHEA@Gel. **k** The photocrosslinking mechanism of GelMA. Cryo-scanning electron microscopy images of (**l**) Gel, (**m**) GHEA@Gel, and (**n**) RHEA@Gel. Scale bars, 20 μm. The insets in (**l**–**n**) represented the photographs of of Gel, GHEA@Gel, and RHEA@Gel, respectively. For these morphological characterizations of the fabricated Gel, GHEA@Gel, and RHEA@Gel, each experiment was repeated three times independently with similar results.

the amount of AST released from HEA protoplasts. In contrast, in HEA with retained cell walls, the rate of AST release significantly decreased after 6 h. This phenomenon may be attributed to the hindrance of further AST release by the presence of cell walls following the release of some easily releasable AST in the earlier stage. Additionally, by collecting the released AST in a centrifuge tube, the color change over time was visually observed (Supplementary Fig. 4b).

GelMAs with positive charges, injectables, and in situ photocuring demonstrated enormous potential for the treatment of irregular wounds in this study. Consequently, the GelMA solution was combined with HEA cells and then coated on the wound; after 5 s of blue light irradiation, the hydrogel was cured and fit perfectly into the wound site. Figure 2j, k show images of pure Gel, GHEA@Gel, and RHEA@Gel, as well as images of the photocuring mechanism. The morphology and structure of the as-prepared Gel and HEA@Gel were subsequently examined via cryo-scanning electron microscopy (FEI Quanta 450). As shown in Fig. 2l, the gel sample exhibited an evidently porous structure. GHEA@Gel and RHEA@Gel retained their polyporous structure, guaranteeing $O_2$ and AST transportation (Fig. 2m, n).

## Antibacterial activity of the GHEA@Gel

Due to the constant exposure of DU wounds to the external environment, there is a high risk of external bacterial infection and significantly delayed wound healing. As previously reported[74,75], different kinds of microorganisms interact with each other to inhibit each other's growth in the case of nonsymbiosis, which contributes to wound antibacterial activity. To verify the nonsyniotic effect of the combination treatment, the proliferation of *Escherichia coli* (*E. coli*) and *Staphylococcus aureus* (*S. aureus*) after cocultivation with GHEA without light irradiation was measured. As shown in Supplementary Fig. 5, obvious inhibition of bacterial growth was observed for both *E. coli* and *S. aureus*, in which the survival rates of *E. coli* and *S. aureus, respectively*, remained at only 48.5% and 46.8%, respectively, when the density of the GHEA cells reached $1 \times 10^8$. In addition, in light of the possibility of photothermal conversion of HEA cells, we further evaluated the in vitro antibacterial activity of GHEA by assessing the photothermal effect under 658 nm laser irradiation (Fig. 3a). As shown in Fig. 3b, the photothermal effect of HEA@Gel was proportional to the HEA cell density. The temperature of the water only increased by 3.5 °C under 658 nm laser irradiation.

When the density of the GHEA cell reached $1 \times 10^8$, the temperature of the GHEA@Gel increased by 85 °C, indicating a robust photothermal effect. *E. coli* and *S. aureus* were subsequently chosen as typical bacteria for investigating the PTT-mediated antibacterial activity of GHEA@Gel. After various processing steps, the optical density (OD) of the bacterial suspension at 600 nm was measured to assess bacterial activity. As shown in Fig. 3c–f, the photothermal effect of GHEA@Gel in combination with laser irradiation significantly inhibited bacterial growth. To further determine the antibacterial activity of the GHEA@Gel, an agar plate test was performed (Fig. 3g). After GHEA@Gel and 658 nm laser irradiation, the number of bacterial colonies on the agar plate was negligible. We hypothesized that the rapid rupture of the bacterial membranes could be the result of a local hyperthermal effect triggered by the powerful photothermal effect of the GHEA@Gel. We then evaluated the integrity of the bacterial membrane using SEM (Fig. 3h). Normal bacteria were observed to have a rhabditiform or spherical morphology with a smooth surface. Following treatment with GHEA@Gel and laser irradiation at 658 nm, the cell membranes of the bacteria were severely compromised, with obvious surface collapse. By destroying the bacterial membrane, GHEA@Gel, which has a 658 nm laser-mediated photothermal effect, was rapidly sterilized. The photothermal conversion of the GHEA@Gel was then evaluated further in vivo. As depicted in Fig. 3i, obvious and excellent photothermal conversion was observed in the DU wounds of mice, where a 658 nm laser could increase the temperature to 55 °C in

5 min. A temperature of 55 °C ensured that the bacteria in the wound were effectively killed, coupled with a nonsymbiosis effect on the HEA cells. However, the disadvantages of this process must also be considered, such as the damage caused by high temperature to normal skin tissue at the wound site. To investigate the damage to normal skin at 55 °C, the skin of the mice was scalded with a constant temperature of electric soldering iron at 55 °C for 5 min. Subsequently, the skin damage at the site of the burn was recorded. The results showed that slight red scald marks were left on the skin of the mice after they were scalded at 55 °C, indicating that 55 °C could cause damage to the skin cells of the mice. After 4 days of burn treatment, the red fluorescence gradually faded (Supplementary Fig. 6), demonstrating that the organism has good self-healing ability at 55 °C on mild scald. To further investigate the damage caused by 55 °C to normal skin cells, the epithelial cells were exposed to 55 °C for 5 min, after which FCM was used to detect cell death. The results showed that, compared with the control treatment, 55 °C caused bearable cell death (Supplementary Fig. 7). In summary, although a high temperature of 55 °C had some slight effects on normal cells while killing infected bacteria, during the actual operation process, because the duration of 55 °C was very short (less than 1 min) and because the body has a strong ability to self-heal 55 °C minor burns, these side effects are completely acceptable in the treatment of wound infection.

## $O_2$ release from the GHEA@Gel

The activity of the GHEA@Gel was initially confirmed by exposing it to red light (658 nm, 0.1 W/cm²) and measuring the soluble oxygen concentration with a microelectrode (Fig. 4a). The soluble oxygen concentration in the GHEA@Gel increased from 0 to more than 8 mg/L within 20 min under adequate light conditions, while in the dark, the soluble oxygen concentration decreased from 8 to 0 mg/L within 20 min (Fig. 4b). These findings indicate that the GHEA in the gel is fully active and capable of photosynthesis and respiration. In a certain range, the oxygen production rate was positively correlated with the GHEA@Gel concentration, and the optimal concentration was $1 \times 10^8$/mL (Fig. 4c). The above oxygen generation experiment was conducted under a light intensity of 0.1 W/cm². To investigate whether GHEA can produce oxygen under laser irradiation at an intensity of 0.5 W/cm², GHEA and GHEA@Gel were subjected to 658 nm laser irradiation at an intensity of 0.5 W/cm², and the changes in oxygen availability in the system were measured every 2 min for a total duration of 30 min. The results, as shown in Supplementary Fig. 8, indicated that the $O_2$ generation curve was very similar to the curve at 0.1 W/cm⁻². To investigate whether the photothermal effect of a high light intensity of 0.5 W/cm⁻² affects the oxygen production capacity of GHEA in GHEA@Gel, GHEA@Gel samples containing GHEA cells at different concentrations were placed under 658 nm laser irradiation for 5 min. Subsequently, after the GHEA temperature decreased to room temperature, a low-intensity laser (0.1 W/cm⁻²) was used to irradiate the GHEA@Gel, after which the generation of oxygen during the irradiation process was measured. As shown in Fig. 4c, although the photothermal effect weakens the rate of oxygen production in the GHEA, a consistent and stable oxygen output can be detected. The ability of GHEA to produce oxygen over a long period of wound healing treatment is critical to its effectiveness. According to the GHEA oxygen-producing process simulated in vitro over a period of 20 days, the ability of GHEA cells to produce oxygen tends to slowly weaken over time during repeated laser irradiation. However, after 20 days of intermittent laser irradiation, the GHEA cells still retained good oxygen production capacity (Supplementary Fig. 9). Multiple cell activities, such as fibroblast proliferation, keratinocyte migration, and endothelial cell differentiation, are concurrently involved in wound healing; therefore, we further examined the effect of GHEA@Gel on human skin fibroblasts (HSFs), human immortalized keratinocytes (HaCaTs), and human umbilical vein endothelial cells (HUVECs). We first discovered that exposure to

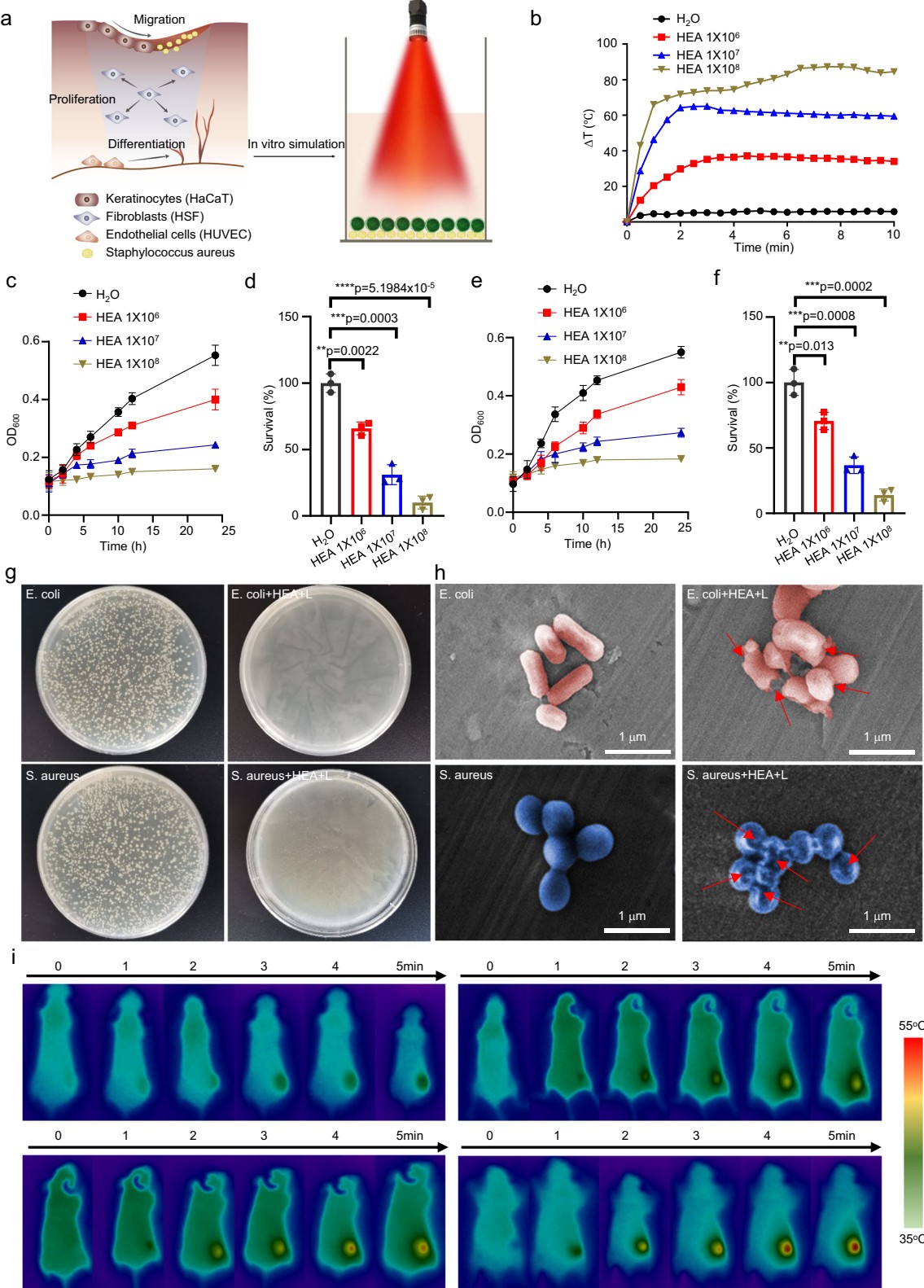

**Fig. 3 | GHEA@Gel photothermal conversion and sterilization. a** Schematic illustration of the photothermal conversion and sterilization of GHEA@Gel. **b** Photothermal transformation of GHEA@Gel. The data are presented as the mean ± s.d. ($n = 3$ independent experiments). **c, d** Quantitative measurement of *E. coli* cells treated with GHEA and a 658 nm laser (0.5 W/cm²). The data are presented as the mean ± s.d. ($n = 5$ biologically independent cells). Statistical differences were analyzed by Student's two-sided *t*-test. **e, f** Quantitative measurement of *S. aureus* cells treated with GHEA and a 658 nm laser (0.5 W/cm²). The data are presented as

the mean ± s.d. ($n = 5$ biologically independent cells). Statistical differences were analyzed by Student's two-sided *t*-test. **g** The corresponding digital images of *S. aureus* and *E. coli* bacterial colonies grown on LB agar plates subjected to various treatments. Each experiment was repeated independently three times with similar results. **h** The corresponding SEM images of *S. aureus* and *E. coli* bacteria subjected to various treatments. Each experiment was repeated independently three times with similar results. **i** Photothermal images of mice receiving various treatments. Each experiment was repeated independently three times with similar results.

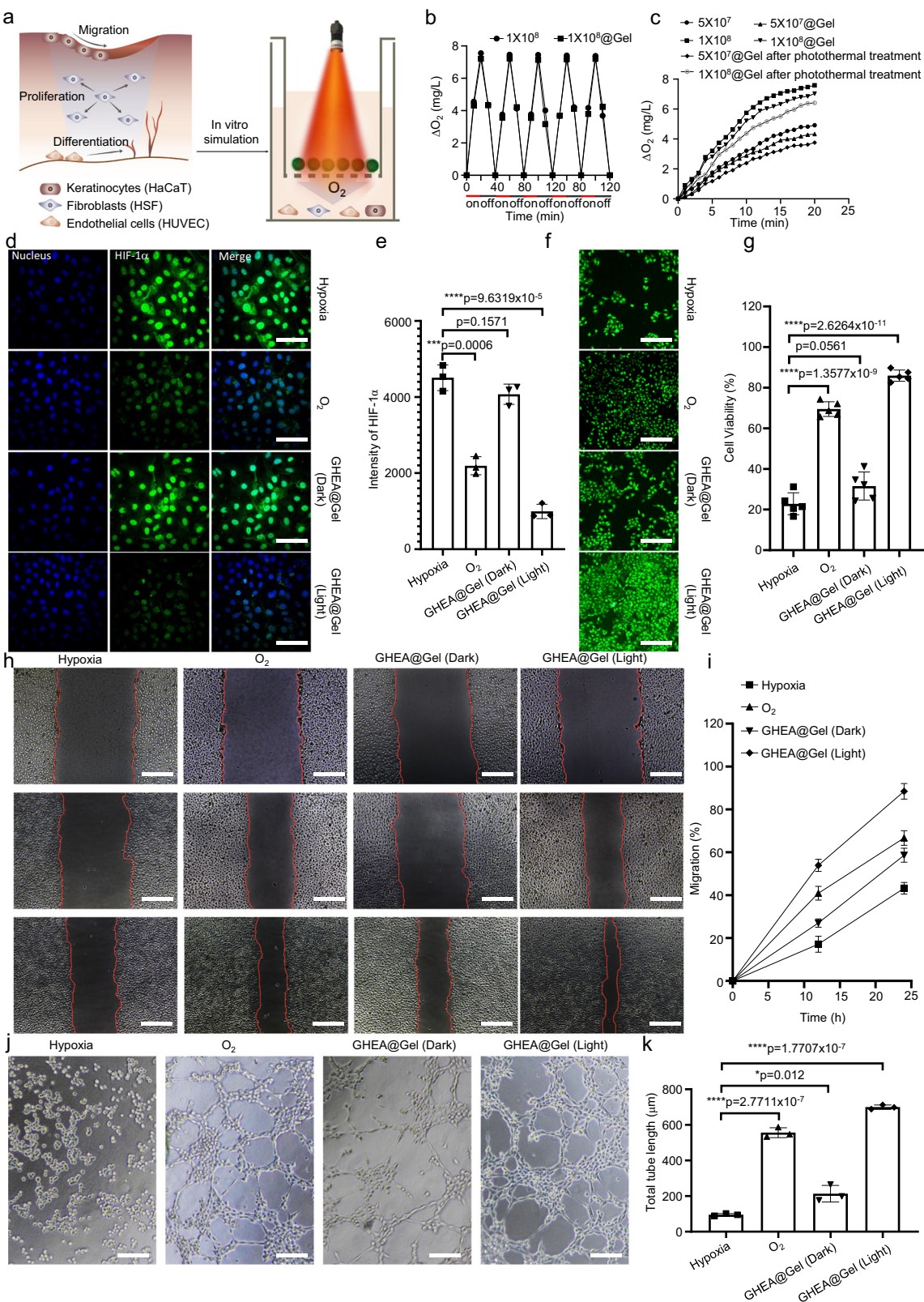

33 mM glucose and 1% $O_2$ induced hypoxia accompanied by over-expression of hypoxia-inducible factor-1α (HIF-1α), indicating the effect of hyperglycemia on hypoxia (Fig. 4d and Supplementary Figs. 10–13). We subsequently determined whether GHEA@Gel could significantly reverse cell hypoxia and result in an 87.9% reduction in HIF-1α in HSFs (Fig. 4e). The same effect was observed for HUVECs and HaCaTs treated with GHEA@Gel (Supplementary Figs. 10–13). In addition to qualitative fluorescence data validation, quantitative analysis via a flow cytometer is desirable. As shown in Supplementary Fig. 14, the flow cytometry data exhibited a trend similar to that of the fluorescence imaging results. The HSFs treated with GHEA@Gel under light conditions exhibited a significantly greater proliferation rate than did those treated with high glucose or even $O_2$ (Fig. 4f, g), indicating the significant potential of GHEA@Gel for forming granulation tissue

**Fig. 4 | O$_2$ release from the GHEA@Gel. a** Schematic representation of O$_2$ release from GHEA@Gel. **b** Comparison of the dissolved O$_2$ release of GHEA@Gel under light and dark conditions. The data are presented as the mean ± s.d. ($n = 3$ independent experiments). **c** The release of dissolved O$_2$ at various GHEA concentrations and laser intensities. **d, e** GHEA@Gel decreased HIF-1α expression in HSFs induced by high glucose concentrations. Scale bars, 100 μm. Each experiment was repeated independently three times with similar results. The data are presented as the mean ± s.d. ($n = 3$ independent cells). Statistical differences were analyzed by Student's two-sided $t$-test. **f, g** Proliferation of HSF cells treated with 33 mM glucose under hypoxia (6 h) for various durations. Scale bars, 200 μm. Each experiment was

repeated independently three times with similar results. The data are presented as the mean ± s.d. ($n = 3$ independent cells). Statistical differences were analyzed by Student's two-sided $t$-test. **h, i** Representative images and quantification of the migration of HaCaT cells. The scale bars are 200 μm. Each experiment was repeated independently three times with similar results. The data are presented as the mean ± s.d. ($n = 3$ independent cells). **j, k** Representative images and quantification of HUVEC tube formation. Scale bars, 200 μm. Each experiment was repeated independently three times with similar results. The data are presented as the mean ± s.d. ($n = 3$ independent cells). Statistical differences were analyzed by Student's two-sided $t$-test.

by releasing dissolved oxygen to promote HSF proliferation. In addition, GHEA@Gel promoted cell proliferation in HUVECs and HaCaTs (Supplementary Figs. 15, 16). Scratch healing of keratinocytes is ideally the core of re-epithelialization processes during chronic wound healing. In the scratch test, we compared cell migration under hypoxic conditions, O$_2$ supply conditions, and soluble oxygen generated by GHEA@Gel under light irradiation. Fig. 4h, i show that, compared with an O$_2$ supply, a hypoxic environment significantly inhibited the migration of HaCaT cells. Notably, coculturing cells with GHEA@Gel in a hypoxic environment and subsequent exposure to light significantly increased the migration rate of the cells, indicating that the soluble oxygen produced by GHEA@Gel could promote epithelialization, thus facilitating chronic wound healing. Endothelial cells are an integral part of blood vessels and play a crucial role in vascularization. We performed tube formation tests using microscopy. The Matrigel tube formation assay, performed to assess angiogenic capacity, revealed a significantly greater tube length and number of branches after 6 h in the GHEA@Gel–treated HUVECs than in the O$_2$-treated HUVECs (Fig. 4j, k). The results of a series of in vitro assays confirmed that the increase in angiogenesis was caused by the soluble oxygen produced by the GHEA@Gel.

## ROS scavenging by RHEA@Gel

Diabetic wounds are characterized by enhanced ROS levels and sustained oxidative stress. The addition of antioxidants to hydrogels has been shown previously to reduce oxidative cellular damage, which speeds up chronic wound healing. Due to the unique molecular structure of AST, RHEA-based hydrogels are promising for promoting diabetic wound healing. AST is the strongest natural antioxidant (Fig. 5a). The clearance of •O$_2^-$, •OH, and H$_2$O$_2$ was subsequently measured to determine the antioxidative performance of RHEA@Gel (Fig. 5b–d). To compare the ROS scavenging ability of RHEA@Gel with greater precision, the natural antioxidant vitamin C (VC) and the chemosynthetic AST were chosen for comparison. As shown in Fig. 5b–d, our prepared RHEA@Gel exhibited the strongest ability to scavenge ROS in the form of •O$_2^-$, •OH, and H$_2$O$_2$ at the same concentration. Using fluorescence microscopy, the ROS scavenging ability of the RHEA@Gel was subsequently confirmed. To simulate the inflammatory microenvironment of diabetic wounds, H$_2$O$_2$ was chosen for further experiments. Specifically, H$_2$O$_2$ solution was added during the HSF cell culture process. Subsequently, after the addition of an ROS scavenger (AST or RHEA@Gel), the cells were incubated for a certain period of time to measure the level of intracellular ROS. Figure 5e, f illustrate that RHEA@Gel is an effective regulator of intracellular ROS levels in HSFs. It is evident from the results that after H$_2$O$_2$ was added to the cell culture, there was a significant increase in the intracellular ROS levels, as indicated by the increase in the intensity of the green fluorescence. However, when AST or RHEA@Gel was added to cells incubated with H$_2$O$_2$, the intracellular ROS concentrations decreased compared with those in the H$_2$O$_2$ group. Additionally, comparable phenomena were observed for HUVECs and HaCaTs (Supplementary Figs. 17, 18). These results indicate that the ability of RHEA@Gel to absorb free radicals is not only significantly greater than that of natural VC but also greater than that of chemically synthesized

AST. The following are the reasons for this outcome. First, AST is a hydrophobic antioxidant that coagulates in aqueous solutions and biological fluids, reducing its bioavailability. Second, the antioxidant activity of AST is related to its molecular structure, and an incorrect molecular structure of chemically synthesized AST also affects its antioxidant activity. Even though the naturally occurring AST produced by RHEA is perfectly formed, its hydrophobicity also affects its antioxidant capacity. In the preliminary characterization, it was determined that loading RHEA protoplasts into the hydrogel generated numerous membrane-wrapped vesicles approximately 100–200 nm in size, similar to exosomes (Fig. 2g, h). The reason why RHEA@Gel has the strongest antioxidant activity is likely due to the precise configuration of naturally secreted AST, and the presence of cell-wrapped exosome vesicles in RHEA@Gel significantly increases the water solubility of AST and enhances its bioavailability.

Based on these findings, the effect of RHEA@Gel on HSF proliferation was further investigated, and RHEA@Gel induced a dramatically greater proliferation rate in HSFs than in H$_2$O$_2$ and AST (Fig. 5g, h), indicating that the AST released from RHEA@Gel has significant potential to form granulation tissue by promoting HSF proliferation. In addition, the ability of RHEA@Gel to promote cell proliferation was also observed for HUVECs and HaCaTs (Supplementary Figs. 19, 20). Next, we examined the influence of AST from RHEA@Gel on ROS-induced HaCaT cells. ROS significantly inhibited HaCaT cell migration, whereas RHEA@Gel stimulation significantly increased HaCaT cell migration (Fig. 5i, j). RHEA@Gel and AST had significant effects on cell migration. We performed tube formation tests using microscopy. The Matrigel tube formation assay, performed to assess angiogenic capacity, revealed a significantly greater tube length and greater number of branches after 6 h in the RHEA@Gel–treated HUVECs than in the AST–treated HUVECs (Fig. 5k, l). In addition, we discovered that the AST released by RHEA@Gel could increase the expression of peroxidase and SOD in cells, thereby scavenging intracellular ROS. These results suggested that the increase in angiogenesis observed in a series of in vitro assays was a result of the release of AST from the RHEA@Gel.

## Anti-inflammatory properties of RHEA@Gel

Macrophages are an important type of immune cell in the immune system that functions by phagocytosis, cytotoxicity, and cytokine secretion. These cells can be activated and polarized into different phenotypes to adapt to various physiological and pathological conditions. M1 macrophages represent a typical proinflammatory phenotype and play a crucial role in antimicrobial and antitumor immunity. The activation of these cytokines is mediated primarily by microbial components (such as lipopolysaccharides) and cytokines (such as IFN-γ). M1 macrophages kill pathogenic microorganisms by producing damaging molecules such as reactive oxygen and nitrogen species and secreting proinflammatory cytokines, thereby triggering local inflammatory responses. M2 macrophages represent a noninflammatory phenotype and are involved primarily in tissue repair, regeneration, and modulation of immune responses. The activation of M2 macrophages is mediated mainly by cytokines (such as interleukin-4 and interleukin-13) and other molecules (such as VEGF and transforming growth factor-beta). M2 macrophages exert their effects by

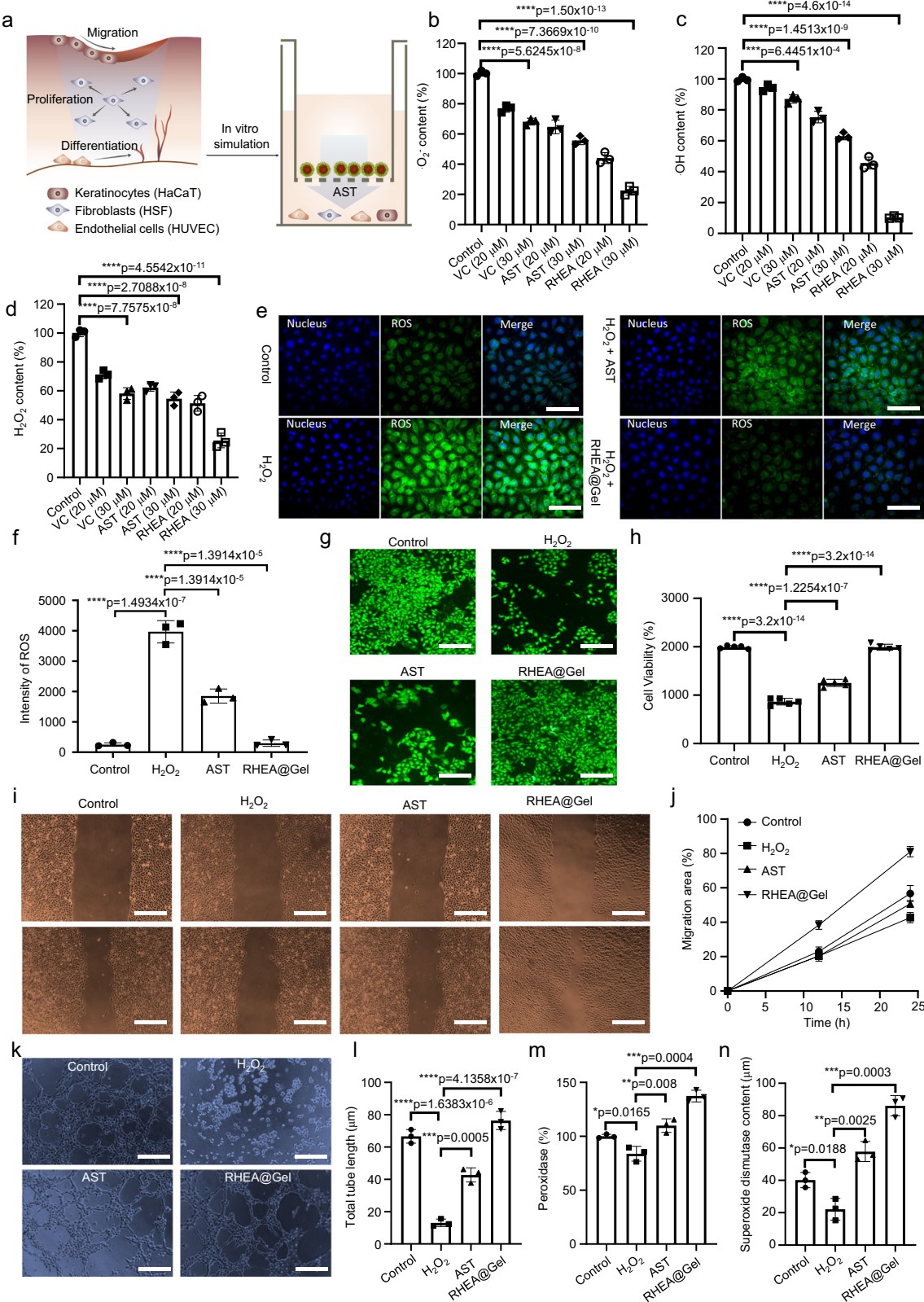

suppressing inflammation, promoting tissue repair, and regulating immune balance. The regulation of these macrophage phenotypes is influenced mainly by various receptors and signaling pathways. For example, Toll-like receptors, nuclear receptors, cytokine receptors, and others can activate specific signaling pathways to regulate macrophage polarization and functions. Different types of pathogens, cellular damage signals, and immune regulatory molecules also

participate in this process, forming a complex signaling network. Depending on the role of different macrophage phenotypes, additional macrophages must be polarized to the M2 phenotype during wound healing. However, the phenotypic switch from M1 to M2 is compromised in diabetic wounds, resulting in sustained inflammation and sluggish wound healing. AST is a natural compound extracted from HEA that has anti-inflammatory properties. We sought to

**Fig. 5 | ROS scavenging ability of RHEA@Gel. a** Schematic representation of ROS scavenging by RHEA@Gel. **b** $O_2^-$, (**c**) OH·, and (**d**) $H_2O_2$-scavenging performance in response to various treatments. The data are presented as the mean ± s.d. ($n = 3$ independent cells). Statistical differences were analyzed by Student's two-sided *t*-test. **e, f** ROS scavenging ability of RHEA@Gel in HSF. Scale bars, 100 µm. Each experiment was repeated independently three times with similar results. The data are presented as the mean ± s.d. ($n = 3$ independent cells). Statistical differences were analyzed by Student's two-sided *t*-test. **g, h** The proliferation of HSF cells treated with 33 mM glucose and $H_2O_2$ in different groups. Scale bars, 200 µm. Each experiment was repeated independently three times with similar results. The data are presented as the mean ± s.d. ($n = 3$ independent cells). Statistical differences

were analyzed by Student's two-sided *t*-test. **i, j** Images and quantification of the migration of HaCaT cells. Scale bars, 200 µm. Each experiment was repeated independently three times with similar results. The data are presented as the mean ± s.d. ($n = 3$ independent cells). **k, l** Representative images and quantitative analysis of tube formation in HUVECs. Scale bars, 200 µm. Each experiment was repeated independently three times with similar results. The data are presented as the mean ± s.d. ($n = 3$ independent cells). **m** Peroxidase and (**n**) superoxide dismutase expression in the different groups. The data are presented as the mean ± s.d. ($n = 3$ independent cells). Statistical differences were analyzed by Student's two-sided *t*-test.

investigate whether the anti-inflammatory effect of AST is due to the regulation of macrophage phenotypic transition and the modification of the immune microenvironment in wounds (Fig. 6a). Briefly, M0 macrophages were stimulated with lipopolysaccharide (LPS) and interferon γ (IFN-γ) to switch to the M1 phenotype as a positive control, while those stimulated with interleukin-4 (IL-4) to switch to the M2 phenotype served as a negative control. CD206 (M2 phenotype) and CD86 (M1 phenotype) macrophages were identified by immuno-fluorescence staining to confirm the polarization status of the macrophages under different stimulation conditions. More CD206+ macrophages and fewer CD86+ macrophages were detected in the GHEA@Gel group than in the control group, indicating that $O_2$ release may have increased M2 activation (Fig. 6b, c and Supplementary Fig. 21). Compared to those in the control and GHEA@Gel groups, macrophages treated with RHEA@Gel displayed significantly more CD206 and less CD86 signaling, indicating that the AST released by RHEA is essential for macrophage polarization. Moreover, the G-RHEA@Gel group displayed the highest CD206 staining and the lowest CD86 signal, further validating that $O_2$ liberation and AST synergistically induced M2 polarization of macrophages. To further validate the ability of HEA to promote the transformation of M1 macrophages into M2 macrophages, flow cytometry was used to detect macrophages in the different treatment groups. The results shown in Fig. 6d were consistent with the confocal microscopy results (Fig. 6d and Supplementary Fig. 22).

### In vivo evaluation of infected diabetic wound healing

The therapeutic approach is depicted in Fig. 7a. Benefiting from the photothermal conversion effect of HEA@Gel under 658 nm laser irradiation at an intensity of 0.5 W/cm$^{-2}$ and the nonsymbiosis effect between HEA and *S. aureus*, the infected diabetic wound was sterilized first. After bactericidal treatment, every subsequent day, a 658 nm laser was used to illuminate the wound at a light intensity of 0.1 W/cm$^{-2}$ for 30 min in the following 10 days. Low-intensity laser irradiation has two effects. First, GHEA activates photosynthesis to produce $O_2$ to improve the hypoxic wound microenvironment and accelerate vascular regeneration and wound healing. The second is to promote the release of AST from RHEA to reduce the levels of overexpressed ROS and inflammatory factors and regulate macrophage M2 polarization in wounds. Hence, HEA@Gel likely promoted fibroblast proliferation, keratinocyte migration, endothelial cell differentiation, and the progressive acceleration of infected wound healing in diabetic patients.

Figure 7b–d shows macroscopic images of the wounds, the wound closure traces at various time points, and the corresponding quantitative wound healing times. On day 3, the wounds in the G-RHEA@Gel with laser group were smaller than those in the other groups. On day 6, the wound area of the G-RHEA@Gel with laser group decreased significantly, whereas the wound sizes in the control, Gel, GHEA@Gel, and RHEA@Gel with laser groups remained at 75%, 72%, 51%, and 44%, respectively. On day 9, 95% of the infected wounds in the group treated with G-RHEA@Gel and the laser had closed. In comparison, the control and gel groups still had substantial wound areas, while the GHEA@Gel and RHEA@Gel groups had faster healing. The

results indicated that treatment with G-RHEA@Gel and laser irradiation promoted the healing of *S. aureus*-infected diabetic wounds.

As depicted in Fig. 7g, histological examination of the healed wound was performed by both hematoxylin-eosin (H&E) and Masson's trichrome (MT) staining. On day 3, none of the groups exhibited epidermal formation or collagen deposition. All the groups exhibited epidermal regeneration on day 9, but the epidermis of the G-RHEA@Gel with laser group was more natural and mature than that of the other groups. As shown in Fig. 7e, the epidermal thicknesses of the control, Gel, GHEA@Gel, RHEA@Gel, and G-RHEA@Gel groups were 115, 110, 90, 60, and 25 µm, respectively. H&E staining demonstrated that infected diabetic wounds in the G-RHEA@Gel group treated with a laser exhibited satisfactory wound healing with accelerated epidermis and dermis formation. In addition, the collagen deposition level on day 9 is shown in Fig. 7f. Collagen deposition was greater in the G-RHEA@Gel with laser group than in the other groups. The H&E and MT staining results suggested that treatment of G-RHEA@Gel with a 658 nm laser could expedite the healing of infected diabetic wounds by promoting the formation of epidermis, facilitating dermal tissue regeneration, and increasing collagen deposition. In addition, immunohistochemical staining using an anti-*Staphylococcus aureus* antibody was applied to assess the bacterial contamination of the wounds. As shown in Fig. 7h, on day 20, the control group and pure gel group wound tissue contained a high number of bacteria (green fluorescence). When HEA@Gel was combined with 658 nm laser irradiation, it was particularly difficult to detect bacteria in the GHEA@Gel, RHEA@Gel, and G-RHEA@Gel groups due to the powerful antibacterial effect of the treatment. The excellent antibacterial effect mediated by HEA could be attributed mainly to the nonsymbiotic effect between bacteria and HEA and the photothermal bactericidal effect of HEA upon irradiation with a 658 nm laser. To further validate this conclusion, the skins of plants subjected to different treatments were collected and homogenized on day 20. Serially diluted homogenates were plated on LB agar to quantify the bacterial colonies. As shown in Supplementary Fig. 23, a large number of *S. aureus* colonies appeared in the control group and the hydrogel group. The number of colonies in the GHEA@Gel group treated with 658 nm laser (0.5 W/cm$^2$) irradiation and the RHEA@Gel group treated with 658 nm laser (0.1 W/cm$^2$) irradiation were greatly reduced due to the photothermal bactericidal effect and nonsymbiotic effect, respectively. Therefore, in view of the above two antibacterial effects, there was almost no colony formation in the G-RHEA@Gel group after treatment.

### Immunofluorescence analysis

To demonstrate the efficacy of HEA@Gel, the ability of GHEA to promote wound healing was examined via HIF-1α immunofluorescence staining. For the control and gel groups, significant HIF-1α expression was noted. Due to the efficient photosynthesis of GHEA, the HIF-1α level in the GHEA@Gel group was significantly lower than that in the control group, indicating a significantly greater oxygen supply (Fig. 8a, e). Compared to those in the GHEA@Gel group, the oxygen supply in the RHEA@Gel group was lower due to the high AST accumulation and reduced photosynthesis. Almost no HIF-1α expression was detected in

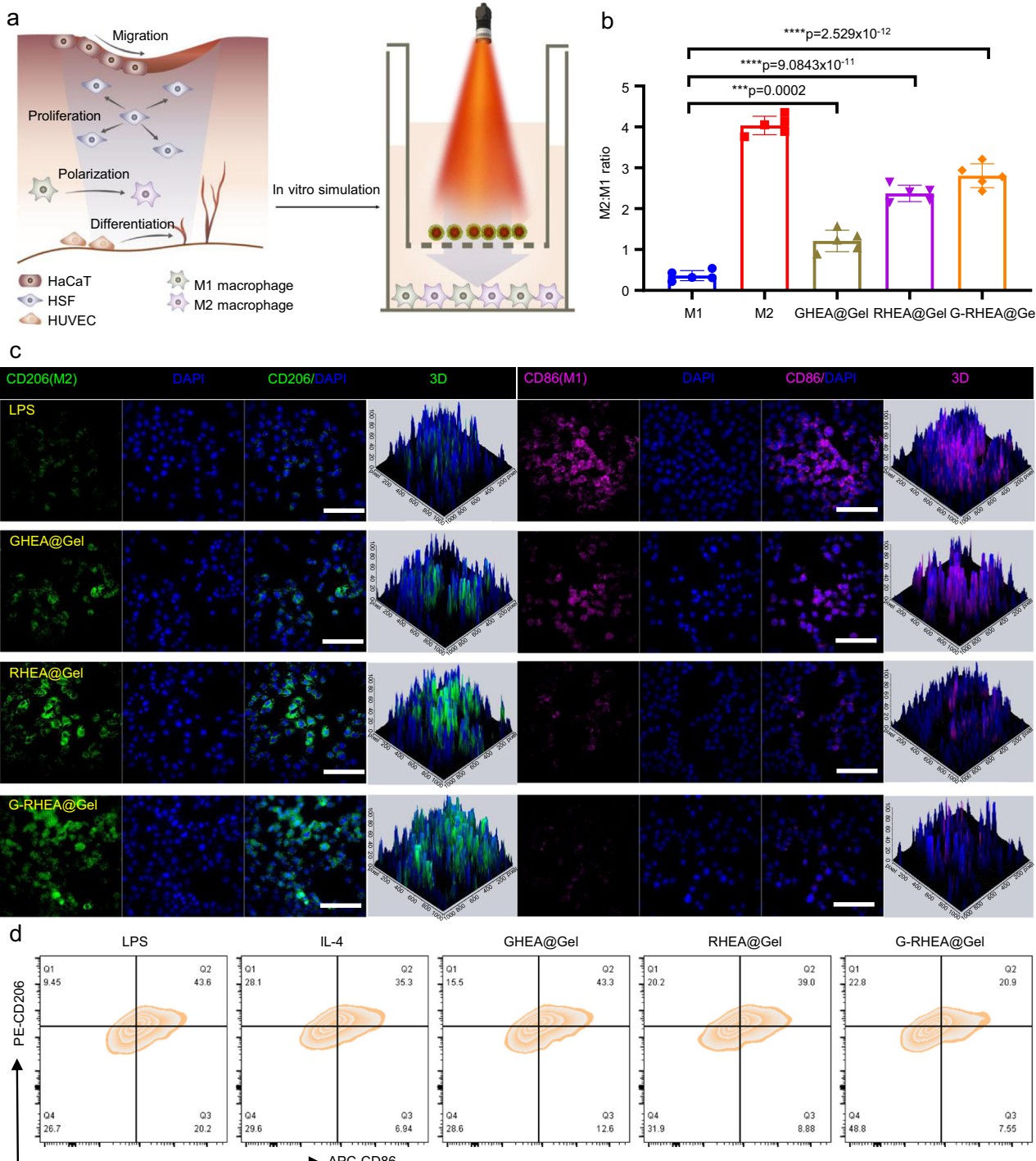

**Fig. 6 | Regulation of macrophage polarization by RHEA@Gel. a** Schematic illustration of the regulation of macrophage polarization by RHEA@Gel. **b** Relative quantification of the ratio of M2 to M1 macrophages. The data are presented as the mean ± s.d. (*n* = 5 independent cells). Statistical differences were analyzed by Student's two-sided *t*-test. **c** Characteristic fluorescence images of Raw264.7 cells stained with CD206 (green) and CD86 (pink) under inflammatory stimulation. Scale bar: 100 μm. **d** Flow cytometry analysis of macrophage polarization after different treatments. Each experiment was repeated independently three times with similar results.

the G-RHEA@Gel group, confirming the high oxygen supply of G-RHEA@Gel. By releasing dissolved oxygen, GHEA@Gel with 658 nm laser irradiation was able to alleviate local hypoxia. To examine the ability of GHEA@Gel to scavenge ROS in vivo, a dihydroethidium (DHE) probe was used to measure the ROS concentration in the wound area. The control and gel groups exhibited bright and high-intensity red fluorescent signals, while the intensity of fluorescence decreased significantly in the GHEA@Gel group, as shown in Fig. 8b, f. In the

RHEA@Gel and G-RHEA@Gel groups, hardly any red fluorescence was detected. According to these results, RHEA@Gel is capable of efficiently scavenging ROS to alleviate oxidative stress.

As depicted in Fig. 8c, g, the underlying biological mechanisms of wound healing were investigated by epidermal growth factor (EGF) immunofluorescence staining on day 6. EGF expression was observed in both the GHEA@Gel and RHEA@Gel groups, indicating that both the oxygen supply capability of GHEA@Gel and the ROS scavenging

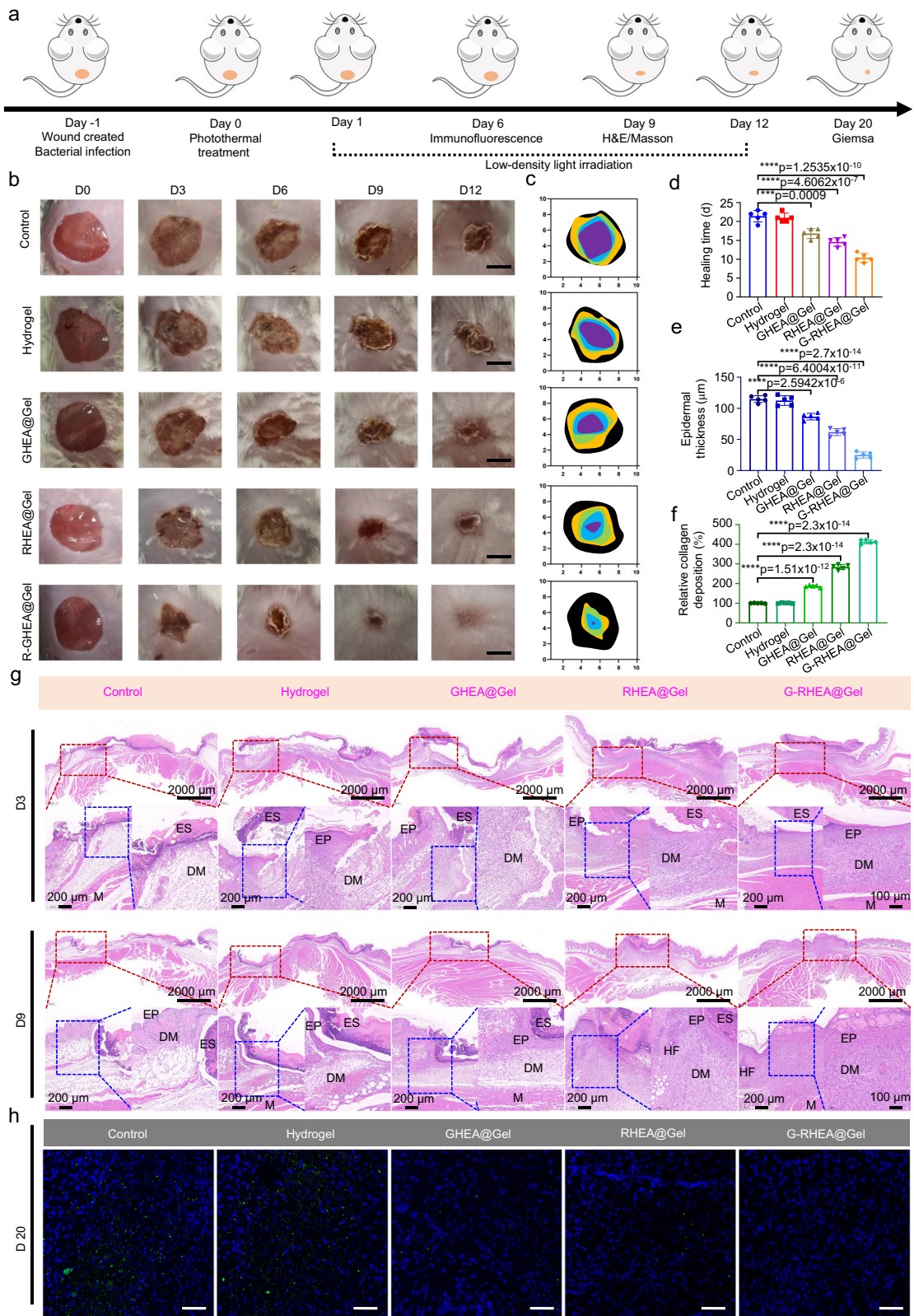

**Fig. 7 | Effects of HEA@Gel on enhancing wound healing in infected diabetic mice. a** Schematic depiction of the sequence of animal experiments conducted to evaluate the therapeutic efficacy of HEA@Gel. **b** Photographs of representative wounds from various treatment groups. **c** Quantitative investigation of the wound area. Each experiment was repeated five times independently with similar results. **d** Statistical analysis of the healing time. The data are presented as the mean ± s.d. (*n* = 5 independent mice). Statistical differences were analyzed by Student's two-sided *t*-test. **e**, **f** Relative quantitative analysis of collagen deposition and epidermal thickness on day 9. The data are presented as the mean ± s.d. (*n* = 5 independent mice). Statistical differences were analyzed by Student's two-sided *t*-test. **g** HE staining on days 3 and 9. Each experiment was repeated five times independently with similar results. **h** Immunohistochemical staining of the wound on day 20. The scale bar is 100 μm. Each experiment was repeated five times independently with similar results.

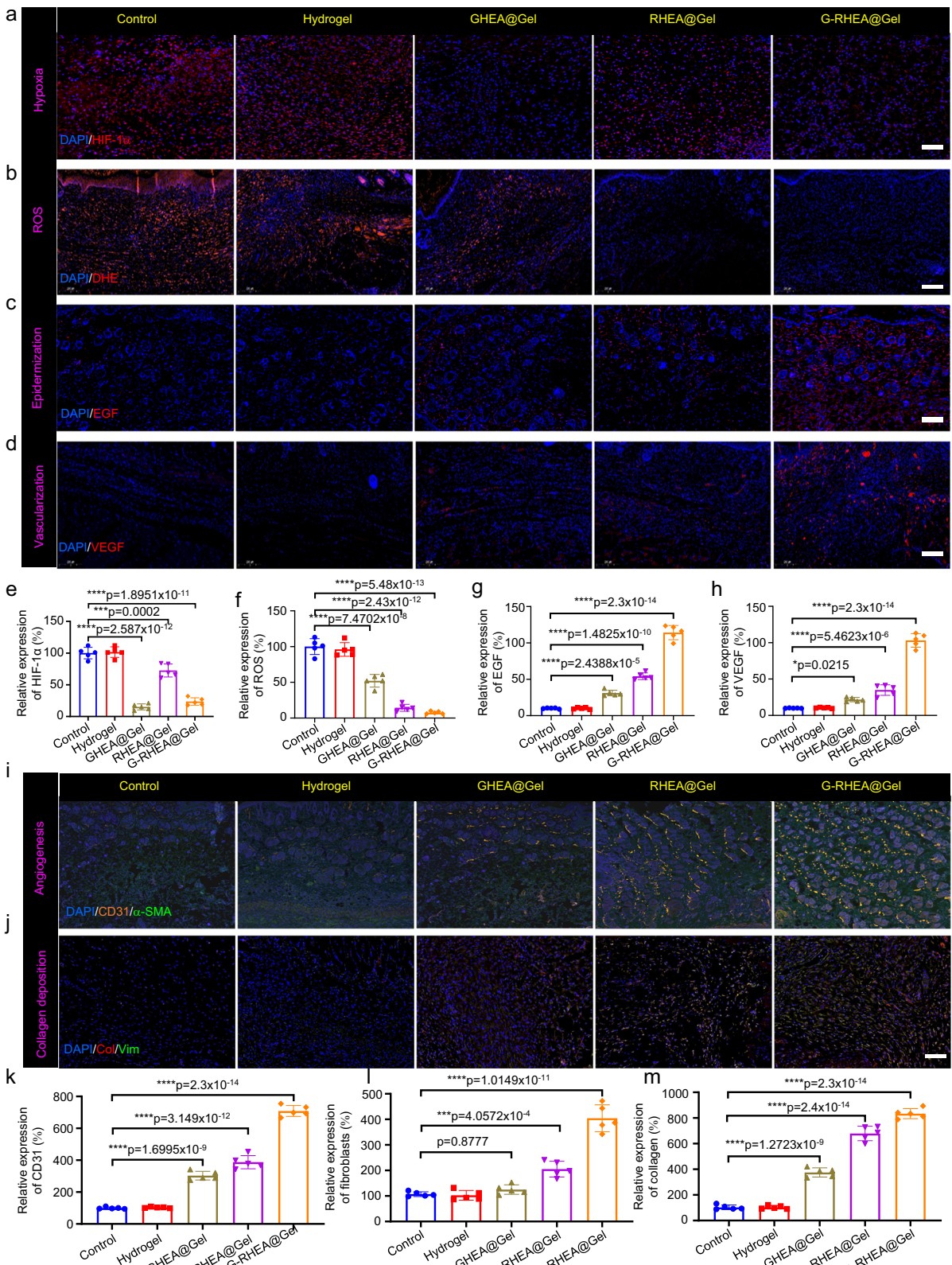

**Fig. 8 | Analysis of the processes and mechanisms of wound healing for various treatments. a** HIF-1α expression, (**b**) ROS content, (**c**) EGF expression, and (**d**) VEGF expression on day 6 under various treatments. The scale bar is 100 μm. Each experiment was repeated independently three times with similar results.
**e**–**h** Quantitative investigation of HIF-1α, ROS, EGF, and VEGF expression on day 6 in response to various treatments. The data are presented as the mean ± s.d. (*n* = 5 independent mice). Statistical differences were analyzed by Student's two-sided *t*-test. Double immunofluorescence staining of (**i**) α-SMA (yellow) and CD31 (green)

and (**j**) the fibroblast markers vimentin (green) and collagen (red) in response to various treatments on day 6. Scale bar, 100 μm. Each experiment was repeated independently three times with similar results. **k**–**m** Quantitative investigation of CD31 expression, neovascularization, and collagen I deposition on day 6 in response to various treatments. The data are presented as the mean ± s.d. (*n* = 5 independent mice). Statistical differences were analyzed by Student's two-sided *t*-test.

capability of RHEA@Gel could promote EGF expression. The highest EGF expression was observed in the G-RHEA@Gel treatment group, indicating that oxygen supply and ROS scavenging are necessary conditions for epidermis formation. In addition, VEGF is an important downstream marker of HIF, and hypoxia suppresses VEGF expression. Figure 8d, h shows the results of VEGF immunofluorescence staining, which revealed a greater level of VEGF in the G-RHEA@Gel group than in the control group, indicating increased blood vessel formation due to the hydrogel, oxygen delivery, and ROS scavenging.

Angiogenesis is a crucial indicator of diabetic wound healing. Neovascularization was evaluated by immunofluorescence staining of a-smooth muscle actin and CD31. As depicted in Fig. 8i, k, the expression of CD31 in the GHEA@Gel and RHEA@Gel groups was significantly greater than that in the control and Gel groups. CD31 expression was highest in the G-RHEA@Gel group. The results demonstrated that G-RHEA@Gel–treated with laser irradiation possessed superior pro-vascularization capacity and promoted wound healing more effectively than the other hydrogels. Therefore, diabetic wound healing can be accelerated by promoting epidermal formation, collagen deposition, and neoangiogenesis. In addition, immunofluorescence staining for collagens and vimentin (a fibroblast marker) was performed to assess collagen deposition in granulation tissue (Fig. 8j, l, and m). ECM formation was facilitated by the increased deposition of matrix and the directional alignment of collagen fibers. The control group had the least amount of collagen, whereas the G-RHEA@Gel group had higher levels of collagen deposition and orientational alignment, suggesting better collagen production favoring wound repair.

### In vivo regulation of macrophage polarization

In the process of wound healing, macrophages play a critical role. The in vivo M2/M1 macrophage ratio was confirmed by immunofluorescence staining for CD206 and CD86 (Fig. 9a). Higher populations of M2 macrophages and lower populations of M1 macrophages were detected in the GHEA@Gel and RHEA@Gel groups, which resulted in a greater M2/M1 macrophage ratio (Supplementary Fig. 24), demonstrating decreased proinflammatory and enhanced anti-inflammatory capacities. In particular, the G-RHEA@Gel group displayed the highest anti-inflammatory capacity, which was attributed to the combined antibacterial, oxygen supply, ROS scavenging, and anti-inflammatory actions of GHEA and RHEA. CD206$^+$ (M2 macrophages) and CD86$^+$ (M1 macrophages) cells were quantified by flow cytometry, and the results were consistent with the immunofluorescence images (Fig. 9b and Supplementary Fig. 25). Nuclear factor-κB (NF-κB) is the central signaling pathway regulating inflammation by mediating the expression of downstream proinflammatory cytokines, including TNF-α, IL-6, and IL-1β, which further impedes tissue recovery. Neutrophils are the predominant effector cells in bacterial infections, and they persistently accumulate in diabetic wounds, thereby retarding the healing process. CXCL-1, a member of the CXC chemokine family, is involved in the migration and activation of neutrophils. At days 3 and 6, the IL-1β, IL-6, TNF-α and CXCL-1 levels in the wounds were reduced in all the gel groups (Fig. 9c–f). The G-RHEA@Gel group exhibited marked decreases in the levels of IL-1β, IL-6, TNF-α and CXCL-1 compared with those in the other groups, indicating the good effectiveness of the combination treatment in alleviating inflammation. In contrast, IL-4 and IL-10 are regeneration-promoting cytokines that facilitate tissue repair, wound healing, axonal regeneration, and M2 macrophage polarization. The G-RHEA@Gel group expressed the highest levels of IL-4 and IL-10 among all the gel groups compared to those in the control group, indicating a remarkable pro-regenerative effect (Fig. 9g, h). On day 3 following treatment, the white blood cell (WBC) count, lymphocyte (Lymph) count, and neutrophil percentage (Gran) in the peripheral blood of the GHEA@Gel, RHEA@Gel, and G-RHEA@Gel groups were significantly lower than those in the control

and hydrogel groups (Fig. 9i–k). Consequently, the synergistic antibacterial, ROS scavenging, oxygenation, and anti-inflammatory effects in the G-RHEA@Gel group could aid in significantly reducing wound inflammation. H&E staining was performed to detect tissue damage, and Supplementary Fig. 26 shows the biosafety of all the hydrogels for in vivo experiments. These results demonstrated that treatment with G-RHEA@Gel and laser stimulation is the optimal wound healing therapy. Furthermore, to assess the safety profile of the hydrogel, G-RHEA@Gel was incubated with HaCaT cells. The results demonstrated excellent safety, as evidenced by the cell activity following coincubation with G-RHEA@Gel. Supplementary Fig. 27a shows that, compared with the control treatment, the G-RHEA@Gel treatment did not adversely affect cell activity. Additionally, staining the cells with calcein-AM and PI dyes and observing them using a confocal laser scanning microscope intuitively revealed that the cells maintained good activity after coincubation with G-RHEA@Gel (Supplementary Fig. 27b). Moreover, the healthy skin of the mice was covered with the G-RHEA@Gel hydrogel for 30 days. Histological examination via H&E and Masson staining revealed no evidence of skin damage following prolonged exposure to G-RHEA@Gel on the skin surface (Supplementary Fig. 28). To further explore the long-term safety of G-RHEA@Gel, the levels of inflammatory factors, including TNF-α, IL-1β, IL-4, IL-6, IL-10, and CXCL-1, and hematological indices, including WBC, lymphocyte, and granule counts, in the peripheral blood of the mice were detected. As shown in Supplementary Fig. 29, there was no significant difference between the G-RHEA@Gel–treated mice and the control group.

## Discussion

Chronic diabetic wounds pose a grave threat to human life and health. The treatment of infected diabetic wounds is a very complex procedure involving bacterial infection, hypoxia, ROS hyperexpression, and inflammation, among other factors. A treatment strategy that is simple and effective is urgently needed. The combination of multifunctional microbial therapies and biocompatible hydrogels holds great promise for treating diabetic wounds. Due to its high efficiency in oxygen production and AST secretion, HEA has emerged as the most valuable microorganism for the treatment of diabetic wounds. However, strategies based on living HEA have not yet been investigated.

In this study, we devised a procedural treatment strategy based on living HEA hydrogels programmed to implement antibacterial, oxygen release, ROS scavenging, anti-inflammatory, and macrophage polarization regulation to promote rapid healing of diabetic wounds in a comprehensive manner. Using simple in situ photocuring technology, HEA@Gel could conform to any asymmetrical wounds. Upon exposure to high-intensity light (658 nm, 0.5 W/cm$^2$), the photothermal conversion of GHEA@Gel had effective antibacterial effects, revealing the initial phase of wound treatment. By regulating low-intensity light (658 nm, 0.5 W/cm$^2$), the GHEA@Gel photosynthetic system can continuously produce oxygen, effectively resolving the problem of hypoxia and promoting vascular regeneration, thereby achieving the objective of the second phase. Continuous light irradiation and the binding effect of the hydrogel network induced AST accumulation in HEA cells, resulting in a gradual change in color from green to red (RHEA). By secreting AST vesicles via exosomes, RHEA is able to efficiently remove excess ROS, enhance the expression of intracellular antioxidant enzymes, and directly polarize M2 macrophages, completing the third phase of infected diabetic wounds. By controlling the light intensity in vitro, the metabolism and function of HEA could be modulated on demand, facilitating programmed wound treatment. Experiments indicate that the living HEA hydrogel can enhance the proliferation and migration of cells, promote tube formation in vitro, and reinforce chronic diabetic wound healing in vivo. Microalgae gel-based treatment provides an advanced strategy for ameliorating chronic wound healing.

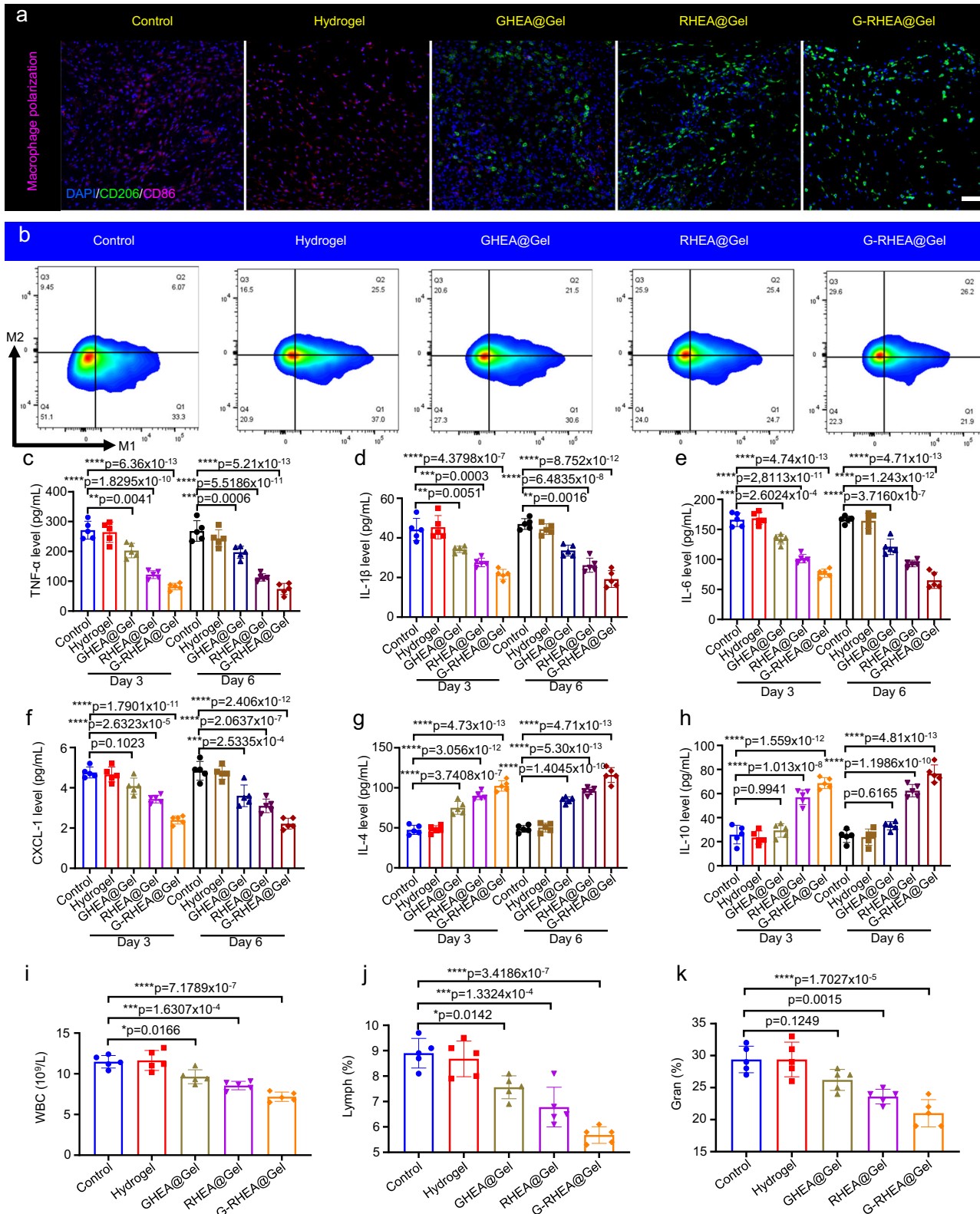

**Fig. 9 | Regulation of macrophage polarization and the immune microenvironment by HEA@Gel in vivo. a** Double immunofluorescence staining of CD206 (green) and CD86 (pink) on day 6 in response to various treatments. Scale bar, 100 μm. Each experiment was repeated independently three times with similar results. **b** Analysis of the ratio of M2 to M1 macrophages in response to various treatments by flow cytometry on day 6. Each experiment was repeated independently three times with similar results. In vivo wound concentrations of (**c**) TNF-α, (**d**) IL-1β, (**e**) IL-6, (**f**) CXCL-1, (**g**) IL-4, and (**h**) IL-10 after 3 and 6 days of treatment. **i** White blood cell count, (**j**) lymphocyte count, and (**k**) neutrophil percentage (gran%) in peripheral blood. The data are presented as the mean ± s.d. (*n* = 5 independent mice). Statistical differences were analyzed by Student's two-sided *t*-test.

Hydrogels loaded with active microalgal cells hold promise as diabetic wound excipients, offering advantages such as enhanced wound healing, antimicrobial properties, and biocompatibility. However, addressing the challenges of limited research, regulatory approval, and cost considerations is crucial for successful clinical transformation. Further research efforts and collaborations between academia, industry, and regulatory bodies are necessary to optimize this innovative approach and facilitate its integration into routine diabetic wound management strategies.

## Methods

### Ethical statement

This research complies with all relevant ethical regulations. All animal studies were conducted in accordance with the National Institute Guide for the Care and Use of Laboratory Animals. All procedures, including animal care, wound modeling, dosing, and termination, were approved by the Tianjin University Institute Animal Ethics Committee with the assigned approval number TJUE-2023-236.

### Materials

HEA cells were supplied by Freshwater Algae Culture Collection at the Institute of Hydrobiology, FACHB. GelMA and photoinitiator LAP were purchased from Suzhou Yongqinquan intelligent equipment Co., LTD. AST (≥97% (HPLC)) were purchased from Solarbio. 2,7-dichlorodihydrofluorescein diacetate (DCFH-DA), and Calcein/PI Cell Viability/Cytotoxicity Assay Kit were purchased from Beyotime. DMEM, RPMI 1640, fetal bovine serum (FBS), trypsin-EDTA, and phosphate-buffered saline (PBS, pH 7.4 and pH 5.5) were supplied by Gibco Life Technologies.

### Preparation and characterization HEA@Gel hydrogel

During the full oscillation period, a certain amount of GelMA and the photoinitiator LAP were dissolved in a certain amount of PBS and mixed uniformly. The mixture was then heated and dissolved at 60 °C for 30 min. The solution was cooled to 37 °C before HEA cells were added and maintained at 37 °C. After 5–10 s of exposure to blue light, the above solution solidified rapidly. The morphology and internal structure of the hydrogels were characterized by cryo-scanning electron microscopy (FEI Quanta 450).

### AST released from RHEA

Five grams of HEA (with cell walls) or HEA protoplast (without cell walls) was separately dispersed in 100 mL of deionized water. The solutions were placed in a room temperature environment and stirred slowly via magnetic stirring. One milliliter was extracted every hour and centrifuged at $1509 \times g$ for 5 min to remove the microalgae, after which the supernatant was collected. The AST in the supernatant was extracted with DMSO, and the content of AST in the supernatant was determined by an AST standard curve. Additionally, to better observe and compare AST release, images of the extracted AST in DMSO solvent were also recorded.

### The photothermal effect of HEA

Different concentrations of GHEA cell solution (1 mL) were exposed to a 658 nm laser (0.5 W/cm²) for 10 min. Using an infrared thermometer (TiS40 Infrared Camera FLUKE-TiS40 9 Hz, FLUKE, USA), the increase in temperature of the GHEA cell solution from photothermal conversion was detected every 30 s for 10 min.

### Antibacterial performance

A plate count strategy was used to evaluate the antibacterial ability of the GHEA. First, 1 mL of *E. coli* or *S. aureus* ($1 \times 10^7$ CFU/mL) was incubated with GHEA@Gel at 37 °C for 12 h. The GHEA@Gel hydrogels were irradiated at 658 nm for 5 min for the laser groups. Then, 0.1 mL of the bacterial suspension was plated on agar and incubated for 24 h at 37 °C. Finally, the number of bacterial colonies was determined, and the antibacterial capacity was determined. To further demonstrate the antibacterial capability of the GHEA, SEM images of the bacterial morphologies under different treatments were examined.

### Extracellular O₂ production

To investigate the oxygen-producing ability of the GHEA cells, a 15 mL final volume of GHEA cell in PBS solution with $5 \times 10^7$ or $1 \times 10^8$ GHEA cells/mL was prepared. Before detection, argon gas was pumped into the solution for 10 min, after which the mixture was left in the dark for 1 h to stabilize the hypoxic conditions. Using an oxygen detector, the light (658 nm, 0.1 W/cm² or 0.5 W/cm²)-assisted GHEA-producing oxygen was measured every minute with stirring. In addition, the solution irradiated by a high-intensity laser (0.5 W/cm²) was left at room temperature for 2 h to cool. The solution was then irradiated with a low-intensity laser (0.1 W/cm²) for 20 min, after which oxygen production was monitored. To determine whether GHEA can produce oxygen during long-term wound healing, a 15 mL final volume of GHEA cell solution with $1 \times 10^8$ GHEA cells/mL was prepared. Then, the solution was irradiated by a high-intensity laser (0.5 W/cm²) for 5 min. Every subsequent day, the solution was irradiated with a low-intensity laser (0.1 W/cm²) for 30 min, and oxygen production was monitored by an oxygen detector.

### ROS scavenging capacity

The ROS scavenging capacity of RHEA@Gel was evaluated using $H_2O_2$, $\cdot O_2^-$, and $\cdot OH$ as representative ROS. The $H_2O_2$ scavenging capacity was determined by mixing 10 mL of 1 mM $H_2O_2$ with the hydrogels. 2 h later, 50 μL of the supernatant was mixed for 30 min with 100 μL of $Ti(SO_4)_2$ solution. The 405 nm absorbance was measured.

The ability to scavenge $\cdot O_2^-$ was evaluated by calculating the inhibition ratio of NBT photoreduction. Under a constant light intensity, NBT (75 μM), methionine (12.5 mM), riboflavin (20 μM), and RHEA@Gel were combined for 10 min. The solution's full wavenumber scanning curve was detected, and the solution's absorbance at 560 nm was recorded.

Using an SA probe, the $\cdot OH$ scavenging ability of RHEA@Gel was determined. Briefly, $FeSO_4$, $H_2O_2$, and SA were added to PBS, and the mixture was incubated for 30 min at 37 °C. The full wavenumber scanning curve of the solution was detected, and the absorbance of the solution at 510 nm was recorded.

### Cell line sources

Human skin fibroblasts (HSF, catalog number: PCS-201-012), mouse monocyte macrophage leukemia cells (RAW264.7, catalog number: TIB-71), human umbilical vein endothelial cells (HUVEC, catalog number: PCS-100-013) and human immortalized keratinocytes (HaCaT, catalog number: PCS-200-011) were obtained from the American Type Culture Collections (ATCC).

### Cell culture

HSFs, HaCaTs, and HUVECs were obtained from ATCC. Dulbecco's modified Eagle's medium (DMEM; Gibco) supplemented with 10% FBS was used to incubate the HSFs. RPMI 1640 medium (Gibco) supplemented with 10% FBS was used to incubate HUVECs, HaCaTs, and RAW264.7.

### Antibodies

Mouse momoclonal Anti-CD86 antibody, Abcam (Product # 130-122-1), Dilution 1:200; Mouse momoclonal Anti-CD206 antibody, Abcam (Product # 141706), Dilution 1:200; Mouse momoclonal Anti-VEGF antibody, Santa Cruzse (Product # 57496), Dilution 1:200; Mouse momoclonal Anti-EGF antibody, Abcam (Product # EPR19173), Dilution 1:500; Mouse momoclonal Anti-HIF-1 alpha antibody, Abcam (Product # EP1215Y), Use a concentration of 0.5 μg/mL; Mouse momoclonal Anti-

CD31 antibody, Abcam (Product # 222783), Dilution 1:1000; Mouse momoclonal Anti-alpha smooth muscle Actin (SMA) antibody, Abcam (Product # ab7817), Dilution 1:500; Mouse momoclonal Anti-VIM antibody, Absin (Product # abs136555), Dilution 1:1000; Mouse momoclonal Anti-COL antibody, Santa Cruzsc (Product # sc-59772), Dilution 1:500; Rabbit polyclonal Anti-Staphylococcus aureus antibody, Absin (Product # ab20920), Dilution 1:1000; Goat Anti-Rabbit IgG (H + L) Highly Cross-Adsorbed Secondary Antibody, ThermoFisher (Catalog # A-11034), Dilution 1:1000; Goat Anti-Mouse IgG (H + L) Highly Cross-Adsorbed Secondary Antibody, ThermoFisher (Catalog # A-21236), Dilution 1:1000.

### Intracellular ROS scavenging determination

The DCFH-DA probe was used to measure the ability of the samples to scavenge ROS. $H_2O_2$ and the samples were added to a 6-well plate containing cultured cells ($5 \times 10^5$ cells/well). The DACH-DA probe was added after 12 h, and the mixture was incubated for another 20 min. Then, fluorescence images were taken for the various groups.

### Intracellular $O_2$ generation

The intracellular oxygen generation capacity of the hydrogels was determined using the $[Ru(dpp)_3]Cl_2$ probe. The cells were cultured in 6-well plate ($5 \times 10^5$ cells/well) and covered with hydrogels, which were either treated with lasers or not. After $[Ru(dpp)_3]Cl_2$ was added to the cells, the cells were incubated for 20 min. Then, fluorescence images were taken for the various groups.

### Cell toxicity and migration

HaCaTs were utilized to assess the cytotoxicity of the hydrogels via an MTT assay. The cells were seeded into the lower chamber of a 24-well transwell plate (each well containing $1 \times 10^5$ cells) for 12 h. Subsequently, the G-RHEA@Gel hydrogel was added to each upper chamber and incubated for another 12 h. Then, MTT solution was added, and the cells were incubated for 2 h. The absorbance at 490 nm was measured via a standardized process. Additionally, calcein-AM ($4 \times 10^{-6}$ M) and PI ($4 \times 10^{-6}$ M) staining were used to distinguish the distribution of living and dead cells. Cell apoptosis was analyzed via confocal laser scanning microscopy. For the cell migration experiment, $1 \times 10^5$ HaCaT cells were seeded in 24-well plates and allowed to grow for 24 h before being scratched. A 100 µL pipette was subsequently used to draw a uniformly wide line in the center of each well of the 24-well plate to remove cells from that area. The cells were then placed in different environments and divided into four groups: the hypoxic environment group, the $O_2$ supply environment group, the hypoxic environment with coincubation with GHEA@Gel group, and the hypoxic environment with coincubation with GHEA@Gel and exposure to light group. Cell migration was recorded at 12 and 24 h thereafter.

### Tube formation assay

To investigate tube formation, 250 µL of thawed Matrigel (abs9493, Absin) was added to each lower chamber of a 24-well plate. After incubation at 37 °C for 1 h, $5 \times 10^4$ HUVECs were seeded into the Matrigel-coated lower chamber and exposed to various treatments. In the laser group, the hydrogel was exposed to 658 nm laser light for 10 min. HUVEC tube images were recorded by a microscope and analyzed using ImageJ software to evaluate tube formation.

### Anti-inflammatory capacity

The anti-inflammatory effect of LPS + IFN-γ or IL-4 was evaluated in a macrophage model. The macrophage was seeded into the lower chamber of a 24-well transwell plate (each well containing $1 \times 10^5$ cells) for 12 h. During macrophage culture, 100 ng/mL LPS and 10 ng/mL IFN-γ were added to the culture medium, after which the cells were cultured for 12 h. Subsequently, GHEA@Gel, RHEA@Gel, or G-RHEA@Gel was added to the culture medium, and the cells were allowed to grow

for 24 h. Macrophages were fixed with 4% paraformaldehyde. Then, immunofluorescence staining and FCM analysis of CD206 and CD86 were performed. During this process, macrophages coincubated with 50 ng/mL IL-4 induced the M2 phenotype as a control.

### Animal study

Female BALB/c mice (6 weeks, 14–16 g) provided by Beijing HFK Bioscience Company were used in this study. All animals were housed in a specific pathogen-free (SPF) animal facility for 2 weeks for environmental adaptation and allowed free access to food and water. During the experiment, all animals were kept in the same standard environment (23–26 °C, 40–60% humidity, 12 h light–dark cycle, and four mice/cage). All procedures, including animal care, wound modeling, dosing, and termination, were performed according to the Experimental Animal Guidelines for Ethical Review of Animal Welfare (GB/T 35892-2018) and approved by the Tianjin University Institute Animal Ethics Committee with the assigned approval number TJUE-2023-236. The mice were anesthetized with isoflurane anesthetic before any procedure that would cause pain. After the experiment, the mice were euthanized by $CO_2$ inhalation followed by cervical dislocation.

### Establishment of an infected diabetic wound model

To evaluate the therapeutic efficacy of HEA@Gel, a mouse model of infected diabetic wounds was created. Female BALB/c mice weighing 25–30 g were fasted for 12 h, followed by intraperitoneal injection of streptozotocin (40 mg/kg), and the process was repeated three times within 3 days. The mice were kept under anesthesia during intraperitoneal injection. All mice were provided with 10% sucrose water. Two weeks after the third injection, diabetic mice were identified as having blood glucose levels exceeding 16.1 mmol/L for two consecutive measurements within 2 days. The diabetic mice were anesthetized with isoflurane anesthetic, and their back fur was completely shaved. Then, 10 mm-diameter skin biopsy punches were used to create full-thickness wounds, and 10 µL of *S. aureus* suspension ($1 \times 10^7$ CFU/mL) was dropped on the surface of the wounds to create an infected wound model. The mice were kept under anesthesia during this procedure. After 1 day of infection, the number of bacterial colonies increased from $10^5$ to $10^6$ CFUs, indicating that the infection model was successfully established.

### Infected diabetic wound healing

After the infected diabetic wound model was established, droplets of hydrogel solution, including hydrogel, GHEA@Gel, RHEA@Gel, or G-RHEA@Gel, were added to the wound. The fluid from the hydrogel was used to quickly cover the entire wound area, after which the hydrogel was irradiated under an ultraviolet lamp for 10 s to achieve hydrogel curing. The mice were kept under anesthesia while the hydrogel was covered and photocuring. The wounds were treated with PBS as a control group. After that, the wounds with different hydrogel covers were exposed to a 658 nm laser with an intensity of 0.5 W/cm$^{-2}$ for 5 min to achieve sterilization through photothermal conversion. The mice were kept under anesthesia during high-power laser (0.5 W/cm$^{-2}$) irradiation. Every subsequent day, the mice were anesthetized with isoflurane anesthetic, and a 658 nm laser was used to illuminate the wound at a light intensity of 0.1 W/cm$^{-2}$ for 30 min in the following 10 days. Low-intensity laser irradiation has two effects. First, GHEA activates photosynthesis to produce oxygen to improve the hypoxic wound microenvironment and accelerate vascular regeneration and wound healing. The second is to promote the release of AST from RHEA to reduce the levels of overexpressed ROS and inflammatory factors and regulate the immune microenvironment in wounds. As the wound healed, the edge of the hydrogel gradually separated from the healed wound tissue, and the hydrogel was slowly removed from the skin on the 12th day after the mice were anesthetized with isoflurane.

On days 0, 3, 6, 9, and 12, wound photographs were taken. On days 3 and 9, the mice were euthanized by $CO_2$ inhalation followed by cervical dislocation, and the skin from the wound site was collected for HE and Masson staining. The hypoxia level, ROS level, vascularization, collagen deposition, epidermis formation, and macrophage polarization of the skin from the wound site were analyzed by immunofluorescence staining for HIF-α, DHE, CD31/α-SMA, collagen/vimentin, EGF, VEGF, and CD206/CD86 on day 6. To evaluate the biocompatibility and biosafety of HEA@Gel-based treatments, the major organs of the mice, including the heart, liver, spleen, lung and kidney, were collected for HE staining analysis on day 8 after HEA@Gel treatment. To evaluate the long-term antibacterial ability of HEA@Gel, the skin at the wound site was collected on day 20, and immunohistochemical staining was performed with an anti-*Staphylococcus aureus* antibody. In addition, to analyze the antibacterial effect visually, the skin was collected and homogenized. Serially diluted homogenates were plated on LB agar to quantify the bacterial colonies.

### Long-term impact of the G-RHEA@Gel on the skin

The safety potential of a material is critical to its application. To explore the long-term impact of HEA@Gel on skin, the healthy skin of female BALB/c mice (6 weeks, 14–16 g) was covered with the G-RHEA@Gel hydrogel for 30 days. Then, the mice were euthanized by $CO_2$ inhalation followed by cervical dislocation, and the skin covered with hydrogel was collected for H&E and Masson staining. In addition, the TNF-α, IL-1β, IL-4, IL-6, IL-10, and CXCL-1 levels and WBC and Lymph and Gran counts were also detected.

### Reporting summary

Further information on research design is available in the Nature Portfolio Reporting Summary linked to this article.

### Data availability

The authors declare that all data supporting the findings of this study are available within the article and the Supplementary Information. All other data are available from the corresponding authors upon request. Source data are provided with this paper.

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

## Acknowledgements

This study was financially supported by a grant from the National Natural Science Foundation of China (Grant No. 32071322, X.J.), National Natural Science Funds for Excellent Young Scholar (Grant No. 32122044, X.J.), China Postdoctoral Science Foundation (Grant No. 2023T160479, 2023M742603, Y.K.), and Technology & Innovation Commission of Shenzhen Municipality (Grant No. JCYJ20210324113004010, X.J.).

## Author contributions

X. Ji, B. Liu, and J. Xie designed and supervised the project. X. Ji, Y. Kang, L. Xu, and J. Dong designed the experimental strategies. Y. Kang, L. Xu, J. Dong, X. Yuan, J. Ye, and Y. Fan performed the experiments and analyzed the data. X. Ji, B. Liu, and J. Xie wrote the manuscript.

## Competing interests

The authors declare no competing interests.
