## [Peer Review File · Nature Communications]

Reviewers' Comments:

Reviewer #1:

Remarks to the Author:

What are the noteworthy results?

This work developed a hydrogel-based treatment strategy with live *Hemococcus* (HEA). By modulating light intensity, HEA can be programmed to perform a variety of functions, such as antibacterial, oxygen supply, and ROS scavenging.

Will the work be of significance to the field and related fields? How does it compare to the established literature? If the work is not original, please provide relevant references. Some previously published articles have used algae containing scaffolds or oxygen release scaffolds to treat diabetic wounds. This work also uses algae (HEA is a type of algae) to release oxygen. The unique contribution of this work is that algae is modulated to have other functions such as antibacterial and ROS scavenging.

Does the work support the conclusions and claims, or is additional evidence needed?
Lots of in vivo data do not support the conclusions and claims.

Are there any flaws in the data analysis, interpretation and conclusions? Do these prohibit publication or require revision?

The quality of images for some in vitro studies and most in vivo studies is poor. It is inappropriate to use these images for quantification and making conclusions. These prohibit publication.

Is the methodology sound? Does the work meet the expected standards in your field?
Yes

Is there enough detail provided in the methods for the work to be reproduced?
No. Methods for the studies are generally vaguely described.

Specific comments:

1. The authors claim that the 3-stage procedure sequentially eliminates bacteria, supplies oxygen, and scavenges ROS. However, bacteria should be continuously eliminated during wound healing, and oxygen supply and ROS scavenging should be performed at the beginning of the treatment. Even when the bacteria are eliminated, the skin cells cannot survive and migrate effectively under the low oxygen and high ROS environment in the diabetic wounds. The details of the treatment are critical. Yet they are missing in the method and results sections. Was the treatment performed only at day 0 as shown in Fig.7a?
2. Fig.2j, chemical structure of the photocrosslinked hydrogel is wrong;
3. Fig.2 k-m, the fabrication condition used cannot result in microporous structure in the hydrogels. This structure is likely artifact, resulting from sample preparation process for SEM imaging;
4. Fig.3i, the use of photothermal effect raised the temperature to 55°C in 5 minutes. Such a high temperature kills not only bacteria but also skin cells. Wound healing would be adversely delayed;
5. The cell migration images in Fig.3h have low quality. The experiment should be conducted in the hypoxia condition since it is used to evaluate the effect of released oxygen on cell migration;
6. What is the concentration of AST released from RHEA? In Fig.5e, the AST control did not show significant ROS scavenging effect. Thus, it is inappropriate to conclude that the AST generated from HEA scavenged ROS;
7. Fig.6, effect of GHEA@Gel and RHEA@Gel on M0 polarization was studied under LPS and IFN- γ stimulation (the condition used for polarizing M0 macrophages into M1 macrophages), it is surprising to see that these macrophages did not turn into M1 macrophages instead of becoming M2 macrophages. Which signaling pathways are responsible for such transition?
8. Fig.9, in vivo macrophage staining has poor quality. CD206 signal looks more like noise. Therefore, the conclusions from in vivo and in vitro studies are inconsistent;
9. Fig.8, in vivo EGF, CD31, α SMA, Vim and Col stainings are in poor quality. The signals are more like noise. It is inappropriate to quantify based on these images;
10. H&E staining images for the Control, Hydrogel and GRHEA@Gel groups in Fig.7g are inconsistent with wound images in Fig.7b. In some images, the quality is poor and there are no

clear epidermis and dermis layers. How did the authors quantify epidermal thickness?

Reviewer #2:

Remarks to the Author:

1. At what point does A produce photothermal or O₂ production and does this require a light source or light intensity? Will O₂ be produced under 658 nm light irradiation at 0.5 W?
2. Does the photothermal effect cause further inflammation when/after killing bacteria?
3. What is the ROS scavenging ability of astaxanthin? How does astaxanthin escape from the algae as well as the hydrogel?
4. Oxygen regulation is one of the main focuses of the article. In addition to qualitative fluorescence data validation, it is desirable to present it in a quantitative (WB or Flow cytometer) manner as well.
5. Where are these hydrogels going to be used in real life? In the article (Introduction and Conclusion), the author can add a description of the flaws and shortcomings of the work.
6. Scheme 1, what is the difference between aminosylated gelatin and regular gelatin? It would be better to have an introduction to the relevant part in the article.
7. In practice, at what time point should the photothermal antibacterial, light-triggered oxygen production and ROS scavenging be performed sequentially?

Reviewer #3:

Remarks to the Author:

In this work, based on the multiple challenges faced by diabetic chronic wounds, such as severe hypoxia, excessive reactive oxygen species (ROS), complex inflammatory microenvironment and potential bacterial infection, the authors developed a programmed microalgae-gel was used to promote the healing of diabetic chronic wounds, and a series of in vitro and in vivo characterizations were performed on the related functions of programmed microalgae-gel in response to the above challenges. However, the design innovation of programmed microalgae-gel is not enough and there are some doubts about the experimental characterization of related functions based on programmed microalgae gel. I think it is not enough to publish in nature communication. The following are some questions and comments about the manuscript work:

1. As the author mentioned, microbes can be a good tool to solve human health problems. For the complex pathological conditions of diabetes, it is reasonable for the authors to adopt an all-in-one microalgae gel as a coping strategy. But similar strategies have been adopted in the reported work, so the design of this programmed microalgae gel is not attractive enough (DOI:10.1016/j.nantod.2021.101368).
2. In the preparation of programmed microalgae gel, the author mentioned that the cell wall of microalgae will affect the release of astaxanthin, but the author did not provide relevant experimental evidence or provide relevant references. The authors talked about using a mild enzymatic method to remove the cell wall, but did not characterize the microalgae activity after cell wall removal by this method. In Figure 2 f, microalgae with cell walls should be provided. Finally, "Interestingly, after a period of culture, HEA protoplasts secreted numerous 200-nm-diameter vesicles, which promoted the effective release and delivery of hydrophobic AST" was described in the author's manuscript, which is not rigorous enough. Because the author has no relevant experiments to demonstrate. the author needs to add experimental evidence for the co-localization of astaxanthins and vesicles.
3. In Figure 3, the authors characterized the antibacterial activity of the programmed microalgae-gel. It is not solid enough for authors to attribute the antimicrobial properties of the programmed microalgae-gel to the photothermal effect. First of all, it has been reported in the literature that

bacteria and algae will inhibit each other in the case of non-symbiosis. Therefore, according to the author's experimental method, the experiments in Figure 3c-f require co-cultivation of microalgae and bacteria without light. Second, the authors provided that the programmed microalgae gel could be heated up to 55°C in vivo, but there was no direct evidence of its antibacterial effect in vivo.

4. In Figure 4, the authors characterize the oxygen production capacity of the programmed microalgae-gel and its function, and the relevant experiments here are sufficiently solid. However, whether the photothermal will affect the oxygen production activity of algae?

5. In Figure 6, the authors characterized the effect of the programmed microalgae gel on the polarization of macrophages through in vitro experiments. The number of M1 and M2 macrophages was quantified by immunofluorescence. Authors should use multiple methods to verify, such as flow cytometry or qPCR.

6. The picture in Figure 7h is not clear enough, the author needs to provide clearer pictures to support the conclusion described by the author. Moreover, the author used an infected wound caused by *S. aureus*, so suggested that using anti-*Staphylococcus aureus* antibody for immunohistochemical staining would be more convincing than Giemsa staining.

Reviewer #1 (Remarks to the Author):

What are the noteworthy results?

This work developed a hydrogel-based treatment strategy with live *Hemococcus* (HEA). By modulating light intensity, HEA can be programmed to perform a variety of functions, such as antibacterial activity, oxygen supply, and ROS scavenging.

Will the work be of significance to the field and related fields? How does it compare to the established literature? If the work is not original, please provide relevant references.

Some previously published articles have used algae-containing scaffolds or oxygen release scaffolds to treat diabetic wounds. This work also uses algae (HEA is a type of algae) to release oxygen. The unique contribution of this work is that algae are modulated to have other functions, such as antibacterial and ROS scavenging.

Does the work support the conclusions and claims, or is additional evidence needed?

Lots of in vivo data do not support the conclusions and claims.

Are there any flaws in the data analysis, interpretation and conclusions? Do these prohibit publication or require revision?

The quality of images for some in vitro studies and most in vivo studies is poor. It is inappropriate to use these images for quantification and making conclusions. These prohibit publication.

Is the methodology sound? Does the work meet the expected standards in your field?

Yes

Is there enough detail provided in the methods for the work to be reproduced?

No. Methods for the studies are generally vaguely described.

Response: We very much appreciate the reviewer's thoughtful and helpful comments. During the past four months, we followed the reviewer's comments and performed additional experiments to address the points raised by the reviewer. Moreover, we have also made a series of modifications/corrections/additions to the manuscript. Please see our point-by-point responses below.

Specific comments:

1. The authors claim that the 3-stage procedure sequentially eliminates bacteria, supplies oxygen, and scavenges ROS. However, bacteria should be continuously eliminated during wound healing, and oxygen supply and ROS scavenging should be performed at the beginning of the treatment. Even when the bacteria are eliminated, the skin cells cannot survive and migrate effectively under the low oxygen and high ROS environment in the diabetic wounds. The details of the treatment are critical. Yet they are missing in the method and results sections. Was the treatment performed only at day 0 as shown in Fig. 7a?

Response: Thank you for your question, and we apologize for the confusion. After establishing the infection model, we used photothermal therapy introduced by a 658 nm laser (0.5 W/cm^2) to kill the bacteria in the wound. The hydrogel material at the wound isolates the direct contact between the wound and the environment and effectively avoids the secondary infection of bacteria to the wound. Subsequently, continuous low-intensity light radiation (0.1 W/cm^2) is carried out during the wound healing period, a process that produces continuous oxygen to promote wound healing. AST also continues to scavenge ROS in the wound during healing. Photothermal sterilization was performed on the first day, and then half an hour of low-intensity light radiation (0.1 W/cm^2) every day was continued until the wound healed. We have supplemented and improved the relevant details about the experimental methods and results in the revised manuscript and have also modified Figure 7a.

Methods section:

Infected Diabetic Wound Healing

After one day of infection, the number of bacterial colonies increased from 10^5 to 10^6 CFU, indicating that the infection model was successfully established. Subsequently, hydrogel, GHEA@Gel, RHEA@Gel, and G-RHEA@Gel evenly covered the wound and were exposed to a 658 nm laser with an intensity of 0.5 W/cm^2 for 5 min. The wound was treated with PBS as a control group. Every subsequent day, a 658 nm laser was used to illuminate the wound at a light intensity of 0.1 W/cm^2 for 30 min until the wound healed.

Figure 7a. Schematic depiction of the sequence of animal experiments conducted to evaluate the therapeutic efficacy of HEA@Gel.

2. Fig.2j, chemical structure of the photocrosslinked hydrogel is wrong;

Response: Thank you for your correction. We apologize for this negligence. The mistake in Figure 2k has been corrected in the revised manuscript.

Figure 2k. The photocrosslinking mechanism of GelMA.

3. Fig. 2 k-m, the fabrication condition used cannot result in microporous structure in the hydrogels. This structure is likely an artifact resulting from the sample preparation process for SEM imaging;

Response: Thank you for your comments, and we apologize for the confusion. The result shown in the picture is the three-dimensional microporous structure of the hydrogel after freeze-drying. In a water environment, the interior of these pores will be filled with water. To better illustrate the internal structure of the hydrogel, we repeated the experiment and provided updated data.

Methods section:

Preparation and Characterization of HEA@GM Hrdrogel

In order to observe the internal structure of the hydrogel, the prepared hydrogel was placed at -80°C for 24 hours to make it fully frozen. The completely frozen hydrogel was then quickly moved to a

freeze dryer and freeze dried for 48 h. The morphology and structure of the cured hydrogels were characterized by SEM.

Results section:

After freeze-drying, the morphology and structure of cured Gel and HEA@Gel were subsequently examined by scanning electron microscopy (SEM). As shown in Figure 2l, the gel sample exhibited an evidently porous structure. GHEA@Gel and RHEA@Gel retained their polyporous structure, guaranteeing O₂ and AST transportation (Figure 2m and 2n).

Figure 2. SEM images of (k) Gel, (l) GHEA@Gel, and (m) RHEA@Gel. For these morphology characterizations of the fabricated Gel, GHEA@Gel, and RHEA@Gel, each experiment was repeated three times independently with similar results.

4. As shown in Fig. 3i, the use of the photothermal effect raised the temperature to 55°C in 5 minutes. Such a high temperature kills not only bacteria but also skin cells. Wound healing would be adversely delayed;

Response: Thank you for your feedback regarding Fig. 3i and the potential adverse effects of the photothermal effect on wound healing. I understand your concern regarding the elevated temperature and its impact on both bacteria and skin cells. When utilizing photothermal therapy, such as raising the temperature to 55°C, it is essential to consider the potential effects on surrounding tissues. While high temperatures can effectively kill bacteria, they can also damage healthy cells, as you mentioned. In our view, bacterial infection of the wound is the first thing to be addressed, and our in vivo experiments show that although high temperature may damage some normal cells, they can effectively eliminate bacterial infection on the wound surface. Compared with nonphotothermal sterilization treatment, the wound healing speed was faster. In addition, in reference to many treatment strategies that induce high temperatures directly on the skin or tissues, a temperature of 55°C does not cause intolerable damage to human tissues. For example, tumor photothermal treatment often needs to induce a high temperature of more than 60°C in the tumor tissue, which is bound to cause damage to the healthy tissues around the tumor, but this damage is inevitable and acceptable. Some relevant statements have been added to our revised manuscript.

Results section:

A temperature of 55°C will ensure that bacteria in the wound are effectively killed. However, the disadvantages of this process must also be considered, such as the damage of high temperature to normal skin tissue of the wound. Further experimental results showed that although a high temperature of 55°C may cause side effects, the wound treated with high temperature still has a good healing speed (Figure 9b), which indicates that the side effect is within the acceptable range.

5. The cell migration images in Fig. 3h have low quality. The experiment should be conducted in the hypoxia condition since it is used evaluate the effect of released oxygen on cell migration;

Response: Thank you for your comment. We conducted the experiment again and updated the low-quality images in Fig. 4h. In the cell migration experiment, we examined the migration of cells

under hypoxic conditions. The relevant results have been explained in more detail in the manuscript.

Methods section:

Cell Toxicity and Migration

HUVECs, HaCaTs, and HSFs were utilized to conduct cell toxicity and cell mobility assays. An MTT assay was performed to determine the hydrogel's toxicity. Briefly, the cells were separated into various treatment groups. Hydrogels were positioned on top of cultured cells. In the laser group, the hydrogel was exposed to 658 nm laser light for 10 min. On day 2, MTT solution was added and incubated for 2 h. The absorbance at 490 nm was measured. For the cell migration experiment, 1×10^5 HaCaT cells were seeded in 24-well plates and allowed to grow for 24 h before being scratched. Then, a 100 μ L pipette was used to draw a uniformly wide line in the center of each well in the 24-well plate to remove cells from that area. The cells were then placed in different environments and divided into four groups: hypoxic environment group, O₂ supply environment group, hypoxic environment with cocultivation with GHEA@Gel group, and hypoxic environment with cocultivation with GHEA@Gel and exposure to light group. Cell migration was recorded at 12 and 24 h thereafter.

Results section:

In the scratch test experiment, we compared cell migration under hypoxic conditions, O₂ supply conditions, and soluble oxygen generated by GHEA@Gel under light irradiation. The results in Figure 4h and 4i show that the hypoxic environment significantly inhibited the migration of HaCaT cells compared to the O₂ supply environment. It is noteworthy that cocultivation of cells with GHEA@Gel in the hypoxic environment and subsequent exposure to light significantly increased the migration rate of cells, indicating that the soluble oxygen produced by GHEA@Gel could promote epithelialization, thus facilitating chronic wound healing.

Figure 4h. Representative images of the migration of HaCaT cells. Scale bars are 200 μ m. Each experiment was repeated independently three times with similar results.

6. What is the concentration of AST released from RHEA? In Fig. 5e, the AST control did not show a significant ROS scavenging effect. Thus, it is inappropriate to conclude that the AST generated from HEA scavenged ROS.

Response: Thank you for your valuable question and suggestion. Regarding the release of AST from HEA, the relevant content has been supplemented in the manuscript. A large number of studies have reported that AST can quench ROS. AST alone did not show a significant ability to reduce ROS in cell experiments, mainly due to the poor water solubility of chemically synthesized AST, which greatly reduced the bioavailability and antioxidant activity of AST. RHEA can release AST in a similar mode of exosome secretion, which greatly increases the water solubility and bioavailability of AST, thus showing stronger ROS quenching ability. The relevant explanation has been added to the article.

Methods section:

AST released from RHEA

Five grams of HEA (with cell wall) and HEA protoplasts (without cell wall) were separately dispersed in 100 mL of deionized water. The solutions were placed in a room temperature environment and stirred slowly using magnetic stirring. One milliliter was extracted every hour and centrifuged at 3000 rpm for 5 min to remove the microalgae, and the supernatant was collected. The AST in the supernatant was extracted with DMSO solvent, and the content of AST in the supernatant was determined by an AST standard curve. Additionally, to better observe and compare AST release, pictures of extracted AST in DMSO solvent were also recorded.

Results section:

To better compare the effects of the presence of the cell wall on AST release, equal amounts of HEA (with the cell wall) and HEA protoplasts (without the cell wall) were separately dispersed in 100 mL of deionized water. The released AST content was measured by using the standard curve of AST in Figure S3. As shown in Figure S4a, the rate of AST release from HEA protoplasts was significantly higher than that of HEA with retained cell walls, and there was a significant difference. Furthermore, with the passage of time, there was a continuous increasing trend in the amount of AST released from HEA protoplasts. In contrast, in HEA with retained cell walls, the rate of AST release significantly decreased after 6 hours. This phenomenon may be attributed to the hindrance of further AST release by the presence of cell walls, following the release of some easily releasable AST in the earlier stage. Additionally, by collecting the released AST in a centrifuge tube, the color change over time can be visually observed (Figure S4b).

Figure S3. (a) Absorbance of AST. (b) Normalized absorbance intensity of AST at 486 nm.

Figure S4. (a) The amount of AST released over time for HEA protoplasts and HEA with cell walls. (b) Photographs of AST solution released at different time points.

Results section:

These results indicate that the ability of RHEA@Gel to absorb free radicals is not only significantly greater than that of natural VC but also greater than that of chemically synthesized AST. The following are the reasons for this outcome. First, AST is a hydrophobic antioxidant that coagulates in aqueous solutions and biological fluids, reducing its bioavailability. Second, the antioxidant activity of AST is related to its molecular structure, and the incorrect molecular structure of chemically synthesized AST also affects its antioxidant activity. Even though the naturally occurring AST produced by RHEA is perfectly formed, its hydrophobicity also affects its antioxidant capacity. In the preliminary characterization, it was determined that loading the RHEA protoplasts into the hydrogel would generate numerous membrane-wrapped vesicles of approximately 100-200 nm in size, similar to exosomes (Figure 2g and 2h). It is theorized that the reason RHEA@Gel has the strongest antioxidant activity is due to the precise configuration of naturally secreted AST and that the cell-wrapped exosome vesicles of RHEA@Gel significantly increase the water solubility of AST and enhance its bioavailability.

7. Fig. 6, effect of GHEA@Gel and RHEA@Gel on M0 polarization was studied under LPS and IFN- γ stimulation (the condition used for polarizing M0 macrophages into M1 macrophages), it is surprising to see that these macrophages did not turn into M1 macrophages instead of becoming M2 macrophages. Which signaling pathways are responsible for this transition?

Response: Thank you for your valuable comment. We apologize for our mistake and the illogical figures in Figure 6. M0 macrophages were stimulated with lipopolysaccharide (LPS) and interferon γ (IFN- γ) to switch to the M1 phenotype as a positive control, while those stimulated with interleukin4 (IL4) to switch to the M2 phenotype served as a negative control. In addition, to further validate the promoting effect of AST on the transformation of M1 macrophages into M2 macrophages, flow cytometry was used to detect macrophages treated with different groups. Inappropriate statements have been corrected, and misleading pictures have been rearranged.

Methods section:

Anti-Inflammatory Capacity

In a macrophage model induced by LPS+IFN- γ or IL4, the anti-inflammatory capacity was evaluated. During macrophage culture, 100 ng/mL LPS and 10 ng/mL IFN- γ were added to the culture medium and cultured for 12 hours. Subsequently, GHEA@Gel, RHEA@Gel, and G-RHEA@Gel were added to the culture medium and continued to grow for 24 hours. Macrophages

were fixed with 4% paraformaldehyde, and immunofluorescence staining for CD206 and CD86 was performed. During this process, macrophages coincubated with IL4 induced the M2 phenotype as a control.

Results section:

Briefly, M0 macrophages were stimulated with lipopolysaccharide (LPS) and interferon γ (IFN- γ) to switch to the M1 phenotype as a positive control, while those stimulated with interleukin 4 (IL4) to switch to the M2 phenotype served as a negative control. CD206 (M2 phenotype) and CD86 (M1 phenotype) macrophages were identified by immunofluorescence staining to confirm the polarization status of macrophages under different stimulation conditions. More CD206 and fewer CD86 macrophages were detected in the GHEA@Gel group than in the control group, indicating that O₂ release may have increased M2 activation (Figure 6b, 6c and Figure S18). Compared to the control and GHEA@Gel groups, macrophages treated with RHEA@Gel displayed significantly more CD206 and less CD86 signal, indicating that the AST released by RHEA is essential for the polarization of macrophages. Moreover, the G-RHEA@Gel group displayed the highest CD206 staining and the lowest CD86 signal, further validating that O₂ liberation and AST synergistically induced M2 polarization of macrophages. To further validate the promoting effect of HEA on the transformation of M1 macrophages into M2 macrophages, flow cytometry was used to detect macrophages in different treatment groups. The results shown in Figure 6d were consistent with the confocal microscopy results.

Figure 6. Regulation of macrophage polarization of RHEA@Gel. (a) Schematic illustration of macrophage polarization regulation by RHEA@Gel. (b) Relative quantitative investigation of the ratio of M2 and M1 macrophages. Data are presented as the mean \pm s.d. ($n = 5$ independent cells). (c) Characteristic fluorescence images of Raw264.7 cells with CD206 (green) and CD86 (pink) staining under inflammatory stimulation. Scale bar, 100 μm . (d) Flow cytometry analysis of macrophage polarization under different treatments. Each experiment was repeated independently three times with similar results.

Figure S18. Characteristic fluorescence images of Raw264.7 cells with CD206 (green) and CD86 (pink) staining under IL-4 stimulation.

8. Fig. 9, in vivo macrophage staining has poor quality. The CD206 signal looks more like noise. Therefore, the conclusions from in vivo and in vitro studies are inconsistent.

Response: Thank you for this comment. The immunofluorescence staining of CD206 (green) and CD86 (pink) with high quality was redetected and added to our revised manuscript.

Figure 9. Regulation of HEA@Gel in vivo macrophage polarization and the immune microenvironment. (a) Double immunofluorescence staining of CD206 (green) and CD86 (pink) on day 6 in response to various treatments. Scale bar, 100 μ m. Each experiment was repeated independently three times with similar results. (b) Analysis of the ratio of M2 to M1 macrophages in response to various treatments by flow cytometry on day 6. Each experiment was repeated independently three times with similar results.

9. Fig. 8, in vivo EGF, CD31, aSMA, Vim and Col stainings are in poor quality. The signals are more similar to noise. It is inappropriate to quantify based on these images;

Response: Thank you for your professional advice. Fig. 8 was redetected and added to our revised manuscript.

Figure 8. Analysis of the processes and mechanisms of wound healing for various treatments. (a) HIF-1 α expression, (b) ROS content, (c) EGF expression, and (d) VEGF expression on day 6 under various treatments. The scale bar is 100 μ m. Each experiment was repeated independently three times with similar results. (e, f, g) Quantitative investigation of HIF-1 α , EGF, and VEGF expression on day 6 in response to various treatments. Data are presented as the mean \pm s.d. (n = 3 independent mice). Double immunofluorescence staining of (h) α -SMA (yellow) and CD 31 (green),

(i) fibroblast marker vimentin (green) and collagen (red) in response to various treatments on day 6. Scale bar, 100 μm . Each experiment was repeated independently three times with similar results. (j-l) Quantitative investigation of CD31, neovascularization, and collagen I deposition on day 6 in response to various treatments. Data are presented as the mean \pm s.d. (n = 3 independent mice).

10. H&E staining images for the Control, Hydrogel and GRHEA@Gel groups in Fig. 7g are inconsistent with wound images in Fig. 7b. In some images, the quality is poor, and there are no clear epidermis and dermis layers. How did the authors quantify epidermal thickness?

Response: Thank you for this comment. We have analyzed the inconsistency between the wound images in Figure 7g and Figure 7b that you mentioned. Because the mouse used for slice analysis in Figure 7g is not the same mouse as the one shown with the wound in Figure 7b, there may be differences in the shape of the wound. However, this does not introduce ambiguity in the analysis of the experimental results. Furthermore, we have conducted repeat experiments for Figure 7g to present higher quality results due to the poor image quality. The epidermal thickness was quantified by ImageJ software. The relevant experimental results have been updated and supplemented in the manuscript.

Figure 7. (g) HE wound staining on days 3 and 9. Each experiment was repeated five times independently with similar results. EP, epidermis; DM, dermis; HF, hair follicle; and M, muscle.

Reviewer #2 (Remarks to the Author):

1. At what point does A produce photothermal or O_2 production and does this require a light source or light intensity? Will O_2 be produced under 658 nm light irradiation at 0.5 W?

Response: Thank you for your question. After establishing the infection model, we used photothermal therapy introduced by a 658 nm laser (0.5 W/cm^2) to kill the bacteria in the wound. The hydrogel material at the wound isolates the direct contact between the wound and the environment and effectively avoids the secondary infection of bacteria to the wound. Subsequently, continuous low-intensity light radiation (0.1 W/cm^2) is carried out during the wound healing period, a process that produces continuous oxygen to promote wound healing. AST also continues to scavenge ROS in the wound during healing. Photothermal sterilization was performed on the first day, and then half an hour of low-intensity light radiation (0.1 W/cm^2) every day was continued until the wound healed. For O_2 generation under 658 nm light irradiation at 0.5 W/cm^2 , we detected

O₂ generation under 658 nm light irradiation at 0.5 W/cm² using a dissolved oxygen analyzer. The O₂ generation curve was detected, which is very similar to the curves at 0.1 W/cm². We have supplemented and improved the relevant details about the experimental methods and results in the revised manuscript and have also modified Figure 7a.

Methods section:

Infected Diabetic Wound Healing

After one day of infection, the number of bacterial colonies increased from 10⁵ to 10⁶ CFU, indicating that the infection model was successfully established. Subsequently, GHEA@Gel, RHEA@Gel, and G-RHEA@Gel evenly covered the wound and were exposed to a 658 nm laser with an intensity of 0.5 W/cm² for 5 min. The wound was treated with PBS as a control group. Every subsequent day, a 658 nm laser was used to illuminate the wound at a light intensity of 0.1 W/cm² for 30 min until the wound healed.

Figure 7a. Schematic depiction of the sequence of animal experiments conducted to evaluate the therapeutic efficacy of HEA@Gel.

Extracellular O₂ Production

To investigate the oxygen-producing ability of GHEA cells, a 15 mL final volume of GHEA cell PBS solution with 5×10^7 or 1×10^8 GHEA cells/mL was prepared. Before detection, argon gas was pumped into the solution for 10 min and left in the dark for one hour to stabilize the hypoxic condition. Using an oxygen detector, the light (658 nm, 0.1 W/cm² or 0.5 W/cm²)-assisted GHEA-producing oxygen was measured every minute with stirring. In addition, the solution irradiated by a high-intensity laser of 0.5 W/cm² was left at room temperature for 2 hours to cool down. The solution was then irradiated with a low-intensity laser of 0.1 W/cm² for 20 minutes, and oxygen production was monitored.

Results section:

To investigate whether GHEA can produce oxygen under laser irradiation with an intensity of 0.5 W/cm², GHEA and GHEA@Gel were subjected to 658 nm laser irradiation with an intensity of 0.5 W/cm², and the oxygen changes in the system were measured every 2 minutes for a total duration of 30 minutes. The results, as shown in Figure S6, indicated that the O₂ generation curve is very similar to the curves at 0.1 W/cm².

Figure S6: The release of dissolved O₂ at various GHEA concentrations under 658 nm laser irradiation with an intensity of 0.5 W/cm².

2. Does the photothermal effect cause further inflammation when/after killing bacteria?

Response: Thank you for your valuable question. I understand your concern regarding the elevated temperature and its impact on both bacteria and skin cells. When utilizing photothermal therapy, such as raising the temperature to 55°C, it is essential to consider the potential effects on surrounding tissues. While high temperatures can effectively kill bacteria, they can also damage healthy cells, as you mentioned. In our view, bacterial infection of the wound is the first thing to be addressed, and our in vivo experiments show that although high temperature may damage some normal cells, they can effectively eliminate bacterial infection on the wound surface. Compared with nonphotothermal sterilization treatment, the wound healing speed was faster. In addition, in reference to many treatment strategies that induce high temperatures directly on the skin or tissues, a temperature of 55°C does not cause intolerable damage to human tissues. For example, tumor photothermal treatment often needs to induce a high temperature of more than 60°C in the tumor tissue, which is bound to cause damage to the healthy tissues around the tumor, but this damage is inevitable and acceptable.

Results section:

A temperature of 55°C will ensure that bacteria in the wound are effectively killed. However, the disadvantages of this process must also be considered, such as the damage of high temperature to normal skin tissue of the wound. Further experimental results showed that although a high temperature of 55°C may cause side effects, the wound treated with high temperature still has a good healing speed (Figure 9b), which indicates that the side effect is within the acceptable range.

3. What is the ROS scavenging ability of astaxanthin? How does astaxanthin escape from the algae as well as the hydrogel?

Response: Thank you for your professional question. Many studies have shown that AST has a variety of enzyme activities (such as ascorbate peroxidase (APX), catalase (CAT), superoxide dismutase (SOD), glutathione peroxidase (GPX), glutathione reductase (GR), peroxiredoxin (PrxR), monodehydroascorbate reductase (MDAR), ferritin, thioredoxin reductase (TrxR), thioredoxin (TRX), glutaredoxin (GRX) and monothiol glutaredoxin (MGRX)), which can effectively scavenge ROS.

Results section:

It has been reported that AST has various ROS scavenging enzyme activities, such as ascorbate peroxidase (APX), catalase (CAT), superoxide dismutase (SOD), glutathione peroxidase (GPX), glutathione reductase (GR), peroxiredoxin (PrxR), monodehydroascorbate reductase (MDAR),

ferritin, thioredoxin reductase (TrxR), thioredoxin (TRX), glutaredoxin (GRX) and monothiol glutaredoxin (MGRX)^{57,58}. In addition, AST can inhibit the expression of inflammatory cytokines (IL-1 β , IL-6, TNF- α and COX-2)⁵⁹.

57. Gwak, Y. et al. Comparative analyses of lipidomes and transcriptomes reveal a concerted action of multiple defensive systems against photooxidative stress in *Hematococcus pluvialis*. *J. Exp. Bot.* 65, 4317-4334 (2014).
58. Hu, C. et al. Transcriptomic analysis unveils survival strategies of autotrophic *Hematococcus pluvialis* against high light stress. *Aquaculture* 513, 734430 (2019).
59. Zhang, X. et al. ROS-triggered self-disintegrating and pH-responsive astaxanthin nanoparticles for regulating the intestinal barrier and colitis. *Biomaterials* 292 (2023).

Regarding the release of AST from HEA, the relevant content has been supplemented in the manuscript. For the release of AST, the existence of the cell wall of HEA would block the effective release of AST. Therefore, we used cellulase and pectinase enzymes to remove the cell wall of HEA and obtain HEA protoplasts, which are more conducive to the release of AST. The released AST can reach the wound site through the porous structure of the hydrogel.

Methods section:

AST released from RHEA

Five grams of HEA (with cell walls) and HEA protoplasts (without cell walls) were separately dispersed in 100 mL of deionized water. The solutions were placed in a room temperature environment and stirred slowly using magnetic stirring. One milliliter was extracted every hour and centrifuged at 3000 rpm for 5 min to remove the microalgae, and the supernatant was collected. The AST in the supernatant was extracted with DMSO solvent, and the content of AST in the supernatant was determined by an AST standard curve. Additionally, to better observe and compare AST release, pictures of extracted AST in DMSO solvent were also recorded.

Results section:

To better compare the effects of the presence of the cell wall on AST release, equal amounts of HEA (with the cell wall) and HEA protoplasts (without the cell wall) were separately dispersed in 100 mL of deionized water, and AST release was studied. The AST content was measured by using the standard curve of AST in Figure S3. As shown in Figure S4a, the rate of AST release from HEA protoplasts was significantly higher than that of HEA with retained cell walls, and there was a significant difference. Furthermore, with the passage of time, there was a continuous increasing trend in the amount of AST released from HEA protoplasts. In contrast, in HEA with retained cell walls, the rate of AST release significantly decreased after 6 hours. This phenomenon may be attributed to the hindrance of further AST release by the presence of cell walls, following the release of some easily releasable AST in the earlier stage. Additionally, by collecting the released AST in a centrifuge tube, the color change over time can be visually observed (Figure S4b).

Figure S3. (a) Absorbance of AST. (b) Normalized absorbance intensity of AST at 486 nm.

Figure S4. (a) The amount of AST released over time for HEA protoplasts and HEA with cell walls. (b) Photographs of AST solution released at different time points.

4. Oxygen regulation is one of the main focuses of the article. In addition to qualitative fluorescence data validation, it is desirable to present it in a quantitative (WB or Flow cytometer) manner as well.

Response: We appreciate the suggestion to present the qualitative fluorescence data in a quantitative manner using flow cytometry. We agree that quantitative measurements would strengthen the study's findings and provide more objective data. We will incorporate this recommendation into our revisions and update the manuscript accordingly.

Results section:

In addition to qualitative fluorescence data validation, it is desirable to present it in a quantitative manner with flow cytometer as well. As shown in Figure S11, the data obtained from flow cytometry detection exhibit a similar trend to the results of fluorescence imaging.

Figure S11: The amount of intracellular HIF-1 α expression in HSF cells after treatment under different conditions and detection by FCM.

5. Where are these hydrogels going to be used in real life? In the article (Introduction and Conclusion), the author can add a description of the flaws and shortcomings of the work.

Response: We greatly appreciate the reviewer's valuable comments. The related content has been added to our revised manuscript.

Introduction section:

Hydrogels are three-dimensional crosslinked polymer networks that can absorb and retain a large amount of water or biological fluids. They have a range of physical and chemical properties that make them suitable for various applications in different fields in real life^{13,14}. Hydrogels have found extensive use in the biomedical field due to their biocompatibility. They can be used as scaffolds for tissue engineering, drug delivery systems, and contact lenses. Hydrogels can also be engineered to respond to external stimuli such as temperature, pH, or light, enabling controlled drug release. In the agricultural field, hydrogels can be incorporated into soil to improve water retention and nutrient availability for plants. They help reduce water usage, prevent soil erosion, and promote plant growth by providing a favorable environment for roots¹⁵. In the environmental remediation field, hydrogels can be utilized for wastewater treatment and environmental cleanup. They can absorb and remove contaminants from water, including heavy metals and organic pollutants.

Clearly, the current design for multifunctional hydrogels entails significant issues, such as complex separation, tedious preparation, low synergistic efficiency, and limited space-time control. Therefore, an urgent need exists for a hydrogel dressing with a simple composition but a procedural therapy strategy¹⁶.

13. Talebian, S. et al. Self-Healing Hydrogels: The Next Paradigm Shift in Tissue Engineering? *Advanced Science* 6 (2019).

14. Liu, X., Inda, M.E., Lai, Y., Lu, T.K. & Zhao, X. Engineered Living Hydrogels. *Adv. Mater.* 34 (2022).
15. Louf, J.-F., Lu, N.B., O'Connell, M.G., Cho, H.J. & Datta, S.S. Under pressure: Hydrogel swelling in a granular medium. *Science Advances* 7 (2021).
16. Xu, Y. et al. Robust and multifunctional natural polyphenolic composites for water remediation. *Materials Horizons* 9, 2496-2517 (2022).

6. Scheme 1, what is the difference between aminosylated gelatin and regular gelatin? It would be better to have an introduction to the relevant part in the article.

Response: Thank you for this constructive comment. Aminosylated gelatin, compared to regular gelatin, contains a large number of amino groups. These amino groups can react with methacrylic anhydride to form gelatin methacryloyl. Gelatin methacryloyl has been demonstrated to possess excellent biocompatibility, cell adhesion properties, and mechanical performance. It is widely applied in tissue engineering, drug delivery, 3D printing, and other fields. The related content has been added to our revised manuscript.

Introduction section:

Gelatin methacryloyl (GelMA) is a dual functionalized gelatin obtained through the reaction between aminosylated gelatin and methacrylic anhydride. The abundant amino groups distributed along the main chain of gelatin provide rich reactive sites for methacrylic anhydride. Methacrylic anhydride, bound to the amino groups, can further react with each other to form three-dimensional structures suitable for cell growth and differentiation in scientific research related to technology. GelMA has been demonstrated to possess excellent biocompatibility, cell adhesion properties, and mechanical performance. It is widely applied in tissue engineering, drug delivery, 3D printing, and other fields^{34, 35}.

34. Liu, B. et al. Hydrogen bonds autonomously powered gelatin methacrylate hydrogels with superelasticity, self-heal and underwater self-adhesion for sutureless skin and stomach surgery and E-skin. *Biomaterials* 171, 83-96 (2018).
35. Kurian, A.G., Singh, R.K., Patel, K.D., Lee, J.-H. & Kim, H.-W. Multifunctional GelMA platforms with nanomaterials for advanced tissue therapeutics. *Bioactive materials* 8, 267-295 (2022).

7. In practice, at what time point should photothermal antibacterial activity, light-triggered oxygen production and ROS scavenging be performed sequentially?

Response: Thank you for your professional question. After establishing the infection model, we used photothermal therapy introduced by a 658 nm laser (0.5 W/cm²) to kill the bacteria in the wound. The hydrogel material at the wound isolates the direct contact between the wound and the environment and effectively avoids the secondary infection of bacteria to the wound. Subsequently, continuous low-intensity light radiation (0.1 W/cm²) is carried out during the wound healing period, a process that produces continuous oxygen to promote wound healing. AST also continues to scavenge ROS in the wound during healing. Photothermal sterilization was performed on the first day, but then half an hour of low-intensity light radiation (0.1 W/cm²) every day continued until the wound healed. We have supplemented and improved the relevant details about the experimental methods and results in the revised manuscript and have also modified Figure 7a.

Methods section:

Infected Diabetic Wound Healing

After one day of infection, the number of bacterial colonies increased from 10⁵ to 10⁶ CFU,

indicating that the infection model was successfully established. Subsequently, hydrogel, GHEA@Gel, RHEA@Gel, and G-RHEA@Gel evenly covered the wound and were exposed to a 658 nm laser with an intensity of 0.5 W/cm^2 for 5 min. The wound was treated with PBS as a control group. Every subsequent day, a 658 nm laser was used to illuminate the wound at a light intensity of 0.1 W/cm^2 for 30 min until the wound healed.

Figure 7a:

Figure 7a. Schematic depiction of the sequence of animal experiments conducted to evaluate the therapeutic efficacy of HEA@Gel.

Reviewer #3 (Remarks to the Author):

In this work, based on the multiple challenges faced by chronic diabetic wounds, such as severe hypoxia, excessive reactive oxygen species (ROS), a complex inflammatory microenvironment and potential bacterial infection, the authors developed a programmed microalgae gel to promote the healing of chronic diabetic wounds, and a series of in vitro and in vivo characterizations were performed on the related functions of the programmed microalgae gel in response to the above challenges. However, the design innovation of programmed microalgae gels is not sufficient, and there are some doubts about the experimental characterization of related functions based on programmed microalgae gels. I think it is not enough to publish in nature communication. The following are some questions and comments about the manuscript work:

Response: We greatly appreciate the reviewer's thoughtful and helpful comments. During the past four months, we performed a series of additional experiments to acquire more significant data. All these data have been added accordingly to the revised manuscript. Moreover, we have also made a series of modifications/corrections/additions to the manuscript. We hope that this revised version can now address all the concerns raised by the respected reviewer and satisfy the high publication standard in *Nature Communication*. Below, please also find our point-by-point responses.

1. As the author mentioned, microbes can be a good tool to solve human health problems. For the complex pathological conditions of diabetes, it is reasonable for the authors to adopt an all-in-one microalgae gel as a coping strategy. However, similar strategies have been adopted in the reported work, so the design of this programmed microalgae gel is not attractive enough (DOI:10.1016/j.nantod.2021.101368).

Response: We greatly appreciate the reviewer's valuable comments. As the reviewer mentioned, for the complex pathological conditions of diabetes, it is reasonable to adopt an all-in-one microalgae gel as a coping strategy. For example, Zhou et al. loaded berberine (BBR, a quorum sensing (QS) inhibitor and antibacterial agent) into the natural living microalga *Spirulina platensis* (SP) to form a bioactive hydrogel (BBR@SP gel) in combination with carboxymethyl chitosan/sodium alginate. Under laser irradiation, the BBR@SP gel could constantly release BBR and O_2 and produce reactive oxygen species, resulting in synergistic QS inhibition against

methicillin-resistant *Staphylococcus aureus* (MRSA) combined with chemo-photodynamic therapy. The BBR@SP gel also suppresses and destroys biofilm formation and downregulates the expression of virulence factors. Although microalgae gel-based wound healing strategies have been reported, the diabetic wound programmed treatment strategy based on HEA reported in our manuscript is very different. In our revised manuscript, the previously reported microalgae gel-based wound healing strategies have been included in the Introduction section, and the innovation of our programmed strategy was also further reframed and emphasized in the Abstract, Introduction, and Discussion sections.

Central innovation:

A programmed diabetic wound treatment strategy employing only live *Hematococcus* (HEA) is reported, in which by modulating light intensity *in vitro*, HEA can be programmed to perform a variety of functions by itself, such as antibacterial activity, oxygen supply, ROS scavenging, and immune regulation.

2. In the preparation of programmed microalgae gel, the author mentioned that the cell wall of microalgae will affect the release of astaxanthin, but the author did not provide relevant experimental evidence or provide relevant references. The authors talked about using a mild enzymatic method to remove the cell wall, but did not characterize the microalgae activity after cell wall removal by this method. In Figure 2f, microalgae with cell walls should be provided. Finally, "Interestingly, after a period of culture, HEA protoplasts secreted numerous 200-nm-diameter vesicles, which promoted the effective release and delivery of hydrophobic AST" was described in the author's manuscript, which is not rigorous enough. Because the author has no relevant experiments to demonstrate. the author needs to add experimental evidence for the colocalization of astaxanthins and vesicles.

Response: Thank you for this comment. We appreciate your suggestions and have taken them into consideration for improving our manuscript. Here is how we have addressed each of your concerns:

Experimental Evidence for Cell Wall Influence

We understand your concern regarding the lack of experimental evidence or references to support the assertion that the cell wall of microalgae affects the release of astaxanthin. To address this, we conducted additional experiments to assess the impact of the cell wall on AST release. The results are now included in the revised manuscript, along with relevant references that support our findings.

Methods section:

AST released from RHEA

Five grams of HEA (with cell walls) and HEA protoplasts (without cell walls) were separately dispersed in 100 mL of deionized water. The solutions were placed in a room temperature environment and stirred slowly using magnetic stirring. One milliliter was extracted every hour and centrifuged at 3000 rpm for 5 min to remove the microalgae, and the supernatant was collected. The AST in the supernatant was extracted with DMSO solvent, and the content of AST in the supernatant was determined by an AST standard curve. Additionally, to better observe and compare AST release, pictures of extracted AST in DMSO solvent were also recorded.

Results section:

The solid and thick cell wall of HEA cells results in their extremely low permeability, which prevents AST from being effectively released from within the cells (Figure 2f)⁶⁸⁻⁷⁰.

68. Huang, W.-C., Liu, H., Sun, W., Xue, C. & Mao, X. Effective Astaxanthin Extraction from Wet *Hematococcus pluvialis* Using Switchable Hydrophilicity Solvents. *ACS Sustainable Chemistry*

& *Engineering* **6**, 1560-1563 (2018).

69. Yang, H.E., Yu, B.S. & Sim, S.J. Enhanced astaxanthin production of *Hematococcus pluvialis* strains induced salt and high light resistance with gamma irradiation. *Bioresour. Technol.* **372**, 128651 (2023).
70. Xu, R., Zhang, L., Yu, W. & Liu, J. A strategy for interfering with the formation of thick cell walls in *Hematococcus pluvialis* by downregulating the mannan synthesis pathway. *Bioresour. Technol.* **362**, 127783 (2022)

To better compare the effects of the presence of the cell wall on AST release, equal amounts of HEA (with the cell wall) and HEA protoplasts (without the cell wall) were separately dispersed in 100 mL of deionized water, and AST release was studied. The AST content was measured by using the standard curve of AST in Figure S3. As shown in Figure S4a, the rate of AST release from HEA protoplasts was significantly higher than that of HEA with cell walls, and there was a significant difference. Furthermore, with the passage of time, there was a continuous increasing trend in the amount of AST released from HEA protoplasts. In contrast, in HEA with retained cell walls, the rate of AST release significantly decreased after 6 hours. This phenomenon may be attributed to the hindrance of further AST release by the presence of cell walls, following the release of some easily releasable AST in the earlier stage. Additionally, by collecting the released AST in a centrifuge tube, the color change over time can be visually observed (Figure S4b).

Figure S3. (a) Absorbance of AST. (b) Normalized absorbance intensity of AST at 486 nm.

Figure S4: (a) The amount of AST released over time for HEA protoplasts and HEA with retained cell walls. (b) Photographs of AST solution released at different time points.

Characterization of microalgal activity and morphology after cell wall removal

You rightly pointed out that we did not characterize the microalgae activity after removing the cell wall using the mild enzymatic method. We apologize for this oversight. In fact, many algal cells, including HEA, retain a certain level of viability even after the removal of the cell wall using relatively gentle methods. Moreover, due to the presence of cellular totipotency, protoplasts can regenerate the cell wall on culture media, which is a common practice in the field of breeding. In the revised manuscript, we have included a detailed characterization of the microalgae activity post cell wall removal.

Results section:

As shown in Figure 2g and S1, HEA protoplasts retain their morphology, structure and activity after their cell walls are removed.

Figure S1. Morphology of HEA protoplasts after incubation for (a) 0 h, (b) 12 h, and (c) 24 h.

Microalgae with cell walls are shown in Figure 2f. For better comparison, microalgae without cell walls are also shown in the revised figure.

Figure 2g. TEM image of HEA protoplast.

Colocalization of Astaxanthins and Vesicles:

In the revised manuscript, we have included new experiments specifically designed to investigate the colocalization of AST and vesicles. These experiments involved fluorescence labeling and confocal microscopy, providing visual evidence supporting our claim.

Results section:

Interestingly, after a period of culture, HEA protoplasts secreted numerous 200 nm diameter vesicles, which promoted the effective release and delivery of hydrophobic AST (Figure 2h and 2i). To demonstrate the presence of AST within secretory vesicles, the vesicles were stained with DiI dye for membrane labeling. Due to the inherent fluorescence of AST, AST and the vesicles were observed under a laser confocal microscope, as shown in Figure S2, revealing clear colocalization of AST and the vesicles.

Figure S2: Laser confocal microscope photographs of the colocalization of AST and vesicles.

3. In Figure 3, the authors characterized the antibacterial activity of the programmed microalgae-gel. It is not solid enough for authors to attribute the antimicrobial properties of the programmed microalgae-gel to the photothermal effect. First, it has been reported in the literature that bacteria and algae will inhibit each other in the case of nonsymbiosis. Therefore, according to the author's experimental method, the experiments in Figure 3c-f require cocultivation of microalgae and bacteria without light. Second, the authors provided that the programmed microalgae gel could be heated up to 55°C *in vivo*, but there was no direct evidence of its antibacterial effect *in vivo*.

Response: Thank you for this valuable comment. The proliferation of *Escherichia coli* (*E. coli*) and *Staphylococcus aureus* (*S. aureus*) after cocultivation with HEA without light irradiation has been added to our revised manuscript. In addition, to detect its antibacterial effect *in vivo*, the infected skins were collected and homogenized on day 20. Serially diluted homogenates were plated on LB agar to quantify the bacterial colonies. Direct evidence of its antibacterial effect *in vivo* has also been added to the revised manuscript. The results showed that the HEA-mediated antibacterial effect should be attributed to the combined effect of photothermal sterilization and nonsymbiotic effects.

Results section:

Due to the constant exposure of the DU wound to the external environment, a high risk of external bacterial infection and significantly delayed wound healing exists. As previously reported^{71,72}, different kinds of microorganisms interact with each other to inhibit each other's growth in the case of nonsymbiosis, which would contribute to wound antibacterial activity. To verify the nonsymbiosis effect, *Escherichia coli* (*E. coli*) and *Staphylococcus aureus* (*S. aureus*) proliferation after cocultivation with GHEA without light irradiation was measured. As shown in Figure S4, an obvious inhabitation of bacterial growth was observed in both *E. coli* and *S. aureus*, in which the survival rates of *E. coli* and *S. aureus* remained only 48.5% and 46.8%, respectively, when the density of GHEA cells reached 1×10^8 .

71. Fulbright, S.P. et al. Bacterial community changes in an industrial algae production system. *Algal Research* 31, 147-156 (2018).
72. Mickalide, H. & Kuehn, S. Higher-Order Interaction between Species Inhibits Bacterial Invasion of a Phototroph-Predator Microbial Community. *Cell Systems* 9, 521-533.e510 (2019).

Figure S5: (a, b) Quantitative measurement of *E. coli* cells treated with GHEA. Data are presented as the mean \pm s.d. ($n = 5$ biologically independent cells). (c, d) Quantitative measurement of *S. aureus* cells treated with GHEA. Data are presented as the mean \pm s.d. ($n = 5$ biologically independent cells).

Antibacterial effect in vivo:

Methods section:

On day 20, the skin was removed to assess the antibacterial effect of HEA@Gel through immunohistochemical staining using an anti-*Staphylococcus aureus* antibody. In addition, to analyze the antibacterial effect visually, the skin was collected and homogenized. Serially diluted homogenates were plated on LB agar to quantify the bacterial colonies.

Results section:

In addition, immunohistochemical staining using an anti-*Staphylococcus aureus* antibody was applied to assess bacterial contamination of wounds. As shown in Figure 7h, on day 20, the control group and pure gel group wound tissue contained a high number of bacteria. When HEA@Gel was combined with 658 nm laser irradiation, it was particularly difficult to detect bacteria in the GHEA@Gel, RHEA@Gel, and G-RHEA@Gel groups due to the treatment's powerful antibacterial effect. The excellent antibacterial effect mediated by HEA should be mainly attributed to the nonsymbiotic effect between bacteria and HEA and the photothermal bactericidal effect of HEA with 658 nm laser irradiation. To further validate this conclusion, the skins under different treatments were collected and homogenized on day 20. Serially diluted homogenates were plated on LB agar to quantify the bacterial colonies. As shown in Figure S19, a large number of *S. aureus* colonies appeared in the control group and the hydrogel group. The number of colonies in the GHEA@Gel group with 658 nm laser (0.5 W/cm^2) irradiation and the RHEA@Gel group with 658 nm laser (0.1 W/cm^2) irradiation were greatly reduced due to the photothermal bactericidal effect and nonsymbiotic effect, respectively. Therefore, in view of the above two antibacterial effects, there was almost no colony formation in the G-RHEA@Gel group after treatment.

Figure 7h. Immunohistochemical staining images using anti-Staphylococcus aureus antibody.

Figure S19. Bacterial colonies of skin extract after different treatments.

4. In Figure 4, the authors characterize the oxygen production capacity of the programmed microalgae-gel and its function, and the relevant experiments here are sufficiently solid. However, whether the photothermal will affect the oxygen production activity of algae is unknown.

Response: Thank you for this professional comment. We appreciate your comment regarding the potential impact of photothermal effects on the oxygen production activity of the programmed microalgae gel. The effect of photothermal treatment on the oxygen production activity of GHEA has been tested and added to our revised manuscript.

Methods section:

In addition, the solution irradiated by a high-intensity laser of 0.5 W/cm^2 was left at room temperature for 2 hours to cool down. The solution was then irradiated with a low-intensity laser of 0.1 W/cm^2 for 20 minutes, and oxygen production was monitored.

Results section:

To investigate whether the photothermal effect under a high light intensity of 0.5 W/cm^2 affects the oxygen production capacity of GHEA in GHEA@Gel, GHEA@Gel samples containing GHEA cells of different concentrations were placed under 658 nm laser irradiation for 5 minutes. Subsequently, after the GHEA temperature dropped to room temperature, a low-intensity laser of 0.1 W/cm^2 was used to irradiate the GHEA@Gel, and the generation of oxygen during the irradiation process was measured. As shown in Figure 4c, although the photothermal effect weakens the rate of oxygen production in the GHEA, a consistent and stable oxygen output can be detected.

Figure 4c. The release of dissolved O_2 at various GHEA concentrations and laser intensities.

5. In Figure 6, the authors characterized the effect of the programmed microalgae gel on the polarization of macrophages through in vitro experiments. The number of M1 and M2 macrophages was quantified by immunofluorescence. Authors should use multiple methods to verify, such as flow

cytometry or qPCR.

Response: We appreciate this helpful comment. To address this concern, we performed additional experiments using flow cytometry to confirm the polarization of macrophages. The results obtained from flow cytometry analysis are now included in revised Figure 6. The combined data provide comprehensive evidence supporting our conclusions regarding macrophage polarization.

Methods section:

Anti-Inflammatory Capacity

In a macrophage model induced by LPS+IFN- γ or IL4, the anti-inflammatory capacity was evaluated. During macrophage culture, 100 ng/mL LPS and 10 ng/mL IFN- γ were added to the culture medium and incubated for 12 hours. Subsequently, three hydrogels, GHEA@Gel, RHEA@Gel, and G-RHEA@Gel, were added to the culture medium and continued to grow for 24 hours. Macrophages were fixed with 4% paraformaldehyde, and immunofluorescence staining for CD206 and CD86 was performed to determine the anti-inflammatory capability of RHEA@Gel. During this process, macrophages coincubated with IL4 induced the M1 phenotype as a control.

Results section:

To further validate the promoting effect of AST on the transformation of M1 macrophages into M2 macrophages, flow cytometry was used to detect macrophages treated with different groups. The results shown in Figure 6d were consistent with the confocal microscopy results.

Figure 4d. Flow cytometry analysis of macrophage polarization under different treatments

6. The picture in Figure 7h is not clear enough, the author needs to provide clearer pictures to support the conclusion described by the author. Moreover, the author used an infected wound caused by *S. aureus*, so suggested that using anti-Staphylococcus aureus antibody for immunohistochemical staining would be more convincing than Giemsa staining.

Response: Thank you for your professional suggestion. We apologize for the unclear image in Figure 7h. As suggested by the reviewer, immunohistochemical staining using an anti-Staphylococcus aureus antibody was carried out. The immunohistochemical staining images have been updated in our revised manuscript.

Methods section:

On day 20, the skin was removed to assess the antibacterial effect of HEA@Gel through immunohistochemical staining using an anti-Staphylococcus aureus antibody. In addition, to analyze the antibacterial effect visually, the skin was collected and homogenized. Serially diluted homogenates were plated on LB agar to quantify the bacterial colonies.

Results section:

In addition, immunohistochemical staining using an anti-Staphylococcus aureus antibody was applied to assess bacterial contamination of wounds. As shown in Figure 7h, on day 20, the control

group and pure gel group wound tissue contained a high number of bacteria. When HEA@Gel was combined with 658 nm laser irradiation, it was particularly difficult to detect bacteria in the GHEA@Gel, RHEA@Gel, and G-RHEA@Gel groups due to the treatment's powerful antibacterial effect. The excellent antibacterial effect mediated by HEA should be mainly attributed to the nonsymbiotic effect between bacteria and HEA and the photothermal bactericidal effect of HEA with 658 nm laser irradiation. To further validate this conclusion, the skins under different treatments were collected and homogenized on day 20. Serially diluted homogenates were plated on LB agar to quantify the bacterial colonies. As shown in Figure S19, a large number of *S. aureus* colonies appeared in the control group and the hydrogel group. The number of colonies in the GHEA@Gel group with 658 nm laser (0.5 W/cm^2) irradiation and the RHEA@Gel group with 658 nm laser (0.1 W/cm^2) irradiation were greatly reduced due to the photothermal bactericidal effect and nonsymbiotic effect, respectively. Therefore, in view of the above two antibacterial effects, there was almost no colony formation in the G-RHEA@Gel group after treatment.

Figure 7h. Immunohistochemical staining images using anti-*Staphylococcus aureus* antibody.

Figure S19. Bacterial colonies of skin extract after different treatments.

We hope that these revisions adequately address your concerns. We appreciate your thorough review and your contribution to improving the quality of our manuscript. If you have any further suggestions or questions, please feel free to let us know.

Reviewers' Comments:

Reviewer #1:

Remarks to the Author:

[Note from the Editor: Reviewer #1 was asked to assess the responses given to Reviewer #1 and the original Reviewer #3.]

The revised manuscript addressed some of the critics of reviewer #1. Yet some critical questions have not been answered correctly.

1. Response to comment #3 of reviewer #1. The authors' response confirms that the microporous structure in the SEM images is not from the as-prepared hydrogels. It is artifact from the SEM sample preparation process. To avoid artifact, the authors should use a cryo-scanning electron microscopy. The current images are misleading because they are from the freeze-dried hydrogels, not from the as-prepared hydrogels used for the animal studies that did not go through the freeze-drying process.
2. Response to comment #4 of reviewer #1. The authors should provide in vitro and in vivo data to demonstrate that 55°C did not cause cell death.
3. Response to comment #5 of reviewer #1. The newly provided cell migration images are not convincing as no individual cells can be seen.
4. Response to comment #9 of reviewer #1. HIF-1 α staining for GHEA@Gel, RHEA@Gel, and G-RHE@Gel groups is not convincing. While quality of the images is improved for the rest of staining, the quantitative results based on the images are inconsistent with the images especially for EGF and VEGF staining. In addition, quantification of ROS expression is missing.
5. Response to comment #1 of reviewer #3: the authors should compare more their wound dressings with the published work by others to better demonstrate the novelty of their work.
6. Response to comment #4 of reviewer #3: photothermal treatment largely decreased the oxygen production in Figure 4c during the first 20 minutes. It is necessary to provide data when the irradiation time is 20 days (duration of wound healing treatment). In their response to comment #1 of reviewer #1, the authors claimed that "continuous low-intensity light radiation (0.1W/cm²) is carried out during the wound healing period."
7. Response to comment #5 of reviewer #3: the flow cytometry results lack a control for M2 macrophages. Therefore, it is not convincing that the developed wound dressings can polarize macrophages towards M2 phenotype.
8. Response to comment #6 of reviewer #3: the results of anti-Staphylococcus aureus antibody staining (Fig.7h) is not convincing as no signal can be seen. This result is also inconsistent with that in Figure S19. Therefore, it is questionable that the wound dressings have anti-bacteria property in vivo.

Reviewer #2:

Remarks to the Author:

The revised manuscript could be publication right now.

Reviewer #1:

1. Response to comment #3 of reviewer #1. The authors' response confirms that the microporous structure in the SEM images is not from the as-prepared hydrogels. It is artifact from the SEM sample preparation process. To avoid artifact, the authors should use a cryo-scanning electron microscopy. The current images are misleading because they are from the freeze-dried hydrogels, not from the as-prepared hydrogels used for the animal studies that did not go through the freeze-drying process.

Response: Thank you for your comments. To characterize the internal structure of the hydrogel more strictly and avoid the interference of artifacts in the experimental results, the internal structure of the hydrogel was observed by cryo-scanning electron microscopy (FEI Quanta 450).

Methods section:

Preparation and Characterization of HEA@Gel Hydrogel

The morphology and internal structure of the hydrogels were characterized by cryo-scanning electron microscopy (FEI Quanta 450).

Results section:

The morphology and structure of the as-prepared Gel and HEA@Gel were subsequently examined by cryo-scanning electron microscopy (FEI Quanta 450). As shown in Figure 2l, the gel sample exhibited an evidently porous structure. GHEA@Gel and RHEA@Gel retained their polyporous structure, guaranteeing O₂ and AST transportation (Figure 2m and 2n).

Figure 2. Cryo-scanning electron microscopy images of (l) Gel, (m) GHEA@Gel, and (n) RHEA@Gel. Scale bars, 20 μm .

2. Response to comment #4 of reviewer #1. The authors should provide in vitro and in vivo data to demonstrate that 55 °C did not cause cell death.

Response: Thank you for your thoughtful and helpful comments. To address the concern regarding cell death caused by a temperature of 55 °C, we have conducted additional experiments and included in vitro and in vivo data in the revised manuscript.

Results section:

A temperature of 55 °C will ensure that bacteria in the wound are effectively killed, coupled with the nonsymbiosis effect with HEA cells. However, the disadvantages of this process must also be considered, such as the damage of high temperature to normal skin tissue of the wound. To investigate the damage of normal skin at 55 °C, the skin of mice was scalded with a constant temperature electric soldering iron at 55 °C for 5 min. Subsequently, the skin damage at the site of the burn was recorded. The results showed that slight red scald marks were left on the skin of mice after being scalded at 55 °C, indicating that 55 °C would cause certain damage to the skin cells of mice. After 4 days of burns, the red mark gradually fades (Figure S6), demonstrating that the organism has a good self-healing ability to 55 °C mild scald. To further investigate the damage of 55 °C to normal skin cells, the epithelial cells were exposed to 55 °C for 5 min, followed by FCM

to detect cell death. The results showed that 55 °C caused bearable cell death compared to the control group (Figure S7). In summary, although the high temperature of 55 °C caused some slight effects on normal cells while killing the wound infection bacteria, in the actual operation process, because the duration of 55 °C is very short, less than 1 min, and the body has a strong self-healing ability to 55 °C minor burns, this side effect is completely acceptable in the treatment process of fighting wound infection.

Figure S6. The self-healing ability to 55 °C minor burns of mice.

Figure S7. Flow cytometry analysis of epithelial cells treated at 55 °C for 5 min.

3. Response to comment #5 of reviewer #1. The newly provided cell migration images are not convincing as no individual cells can be seen.

Response: Thank you for your correction. To obtain more convincing experimental results, we reconducted this part of the experiment and provided high-quality images of the experimental results.

Figure 4h. Representative images of the migration of HaCaT cells. Scale bars are 250 μm .

4. Response to comment #9 of reviewer #1. HIF-1 α staining for GHEA@Gel, RHEA@Gel, and G-RHE@Gel groups is not convincing. While the quality of the images is improved for the rest of staining, the quantitative results based on the images are inconsistent with the images, especially for EGF and VEGF staining. In addition, quantification of ROS expression is missing.

Response: We appreciate your comments regarding HIF-1 α staining. We have taken your comments into consideration and made efforts to improve the quality of the images. The HIF-1 α staining experiment was repeated to make the results more convincing. Regarding the quantitative results of EGF and VEGF staining, we apologize for this negligence. The relevant results have been adjusted. Additionally, we apologize for the omission of ROS expression quantification. We have included this part of the experimental data in the revised manuscript.

Figure 8a. HIF-1 α expression on day 6 under various treatments. The scale bar is 100 μm .

Figure 8e, f, g, h. Quantitative investigation of HIF-1 α , ROS, EGF, and VEGF expression on day 6 in response to various treatments.

5. Response to comment #1 of reviewer #3: the authors should compare more their wound dressings

with the published work by others to better demonstrate the novelty of their work.

Response: Thank you for your valuable suggestion. We agree that such a comparison would further enhance the novelty and significance of our work. In response to this suggestion, we have conducted a thorough literature review and included a comprehensive comparison of our wound dressings with the published work of other researchers in the revised manuscript.

Introduction section:

For example, Zhao et al. developed a therapeutic wound dressing, namely, MnCoO@PDA/CPH, utilizing a biomimetic hydrogel system and modified hydrogen peroxide-mimicking nanozymes. The hydrogel is engineered to simultaneously match the mechanical and electrical signals of the skin while possessing oxidative capability activated by H₂O₂³⁴. Wu et al. prepared a versatile dynamic Schiff base and borate ester cross-linked glycopeptide hydrogel that could continuously generate oxygen, promote M2 polarization of macrophages, and eliminate reactive oxygen and nitrogen species³⁵. Zhang et al. prepared an injectable hydrogel based on platelet-rich plasma and laponite that could accelerate wound healing by promoting macrophage polarization and angiogenesis in full-thickness skin³⁶. Clearly, the current design for multifunctional hydrogels entails significant issues, such as complex separation, tedious preparation, low synergistic efficiency, and limited space-time control. Therefore, an urgent need exists for a hydrogel dressing with a simple composition but a procedural therapy strategy. Gelatin methacryloyl (GelMA) is a dual functionalized gelatin obtained through the reaction between aminosylated gelatin and methacrylic anhydride. The abundant amino groups distributed along the main chain of gelatin provide rich reactive sites for methacrylic anhydride. Methacrylic anhydride, bound to the amino groups, can further react with each other to form three-dimensional structures suitable for cell growth and differentiation in scientific research related to technology. GelMA has been demonstrated to possess excellent biocompatibility, cell adhesion properties, and mechanical performance. It is widely applied in tissue engineering, drug delivery, 3D printing, and other fields^{37,38}.

- 34 Zhao, Y. et al. Biomimetic Nanozyme-Decorated Hydrogels with H₂O₂-Activated Oxygenation for Modulating Immune Microenvironment in Diabetic Wound. *ACS Nano* 17, 16854-16869 (2023).
35. Wu, Y. et al. A Versatile Glycopeptide Hydrogel Promotes Chronic Refractory Wound Healing Through Bacterial Elimination, Sustained Oxygenation, Immunoregulation, and Neovascularization. *Adv. Funct. Mater.* 2305992 (2023).
36. Zhang, J. et al. An injectable bioactive dressing based on platelet-rich plasma and nanoclay: Sustained release of deferoxamine to accelerate chronic wound healing. *Acta Pharm. Sin. B* 13, 4318-4336 (2023).
37. Liu, B. et al. Hydrogen bonds autonomously powered gelatin methacrylate hydrogels with super-elasticity, self-heal and underwater self-adhesion for sutureless skin and stomach surgery and E-skin. *Biomaterials* 171, 83-96 (2018).
38. Kurian, A.G., Singh, R.K., Patel, K.D., Lee, J.-H. & Kim, H.-W. Multifunctional GelMA platforms with nanomaterials for advanced tissue therapeutics. *Bioactive materials* 8, 267-295 (2022).

For example, Zhou et al. loaded berberine (BBR, a quorum sensing (QS) inhibitor and antibacterial agent) into the natural living microalga *Spirulina platensis* (SP) to form a bioactive hydrogel (BBR@SP gel) in combination with carboxymethyl chitosan/sodium alginate. Under laser irradiation, the BBR@SP gel could constantly release BBR and O₂ and produce reactive oxygen

species, resulting in synergistic QS inhibition against methicillin-resistant *Staphylococcus aureus* (MRSA) combined with chemo-photodynamic therapy. The BBR@SP gel also suppresses and destroys biofilm formation and downregulates the expression of virulence factors⁵⁵.

55. Hu, H. et al. Microalgae-based bioactive hydrogel loaded with quorum sensing inhibitor promotes infected wound healing. *Nano Today* 42, 101368 (2022).

6. Response to comment #4 of reviewer #3: photothermal treatment largely decreased the oxygen production in Figure 4c during the first 20 minutes. It is necessary to provide data when the irradiation time is 20 days (duration of wound healing treatment). In their response to comment #1 of reviewer #1, the authors claimed that “continuous low-intensity light radiation (0.1 W/cm²) is carried out during the wound healing period.”

Response: Thank you for your comments, and we apologize for the confusion. To determine whether GHEA has the ability to produce oxygen during the long duration of wound healing treatment, the oxygen-producing process of GHEA during wound treatment was simulated in vitro. A certain amount of GHEA was distributed in the solution, and the solution was irradiated by a high-intensity laser of 0.5 W/cm² for 5 min. Every subsequent day, the solution was irradiated with a low-intensity laser of 0.1 W/cm² for 30 min, and oxygen production was monitored.

Methods section:

Extracellular O₂ Production

To determine whether GHEA has the ability to produce oxygen during the long duration of wound healing treatment, a 15 mL final volume of GHEA cell solution with 1×10^8 GHEA cells/mL was prepared. Then, the solution was irradiated by a high-intensity laser of 0.5 W/cm² for 5 min. Every subsequent day, the solution was irradiated with a low-intensity laser of 0.1 W/cm² for 30 min, and oxygen production was monitored by an oxygen detector.

Results section:

The ability of GHEA to produce oxygen over a long period of wound healing treatment is critical to the effectiveness of treatment. According to the GHEA oxygen-producing process simulated in vitro over a period of 20 days, the ability of GHEA cells to produce oxygen has a tendency to slowly weaken over time during repeated laser irradiation. However, after 20 days of intermittent laser irradiation, the GHEA cells still retained good oxygen production capacity (Figure S9).

Figure S9. The release of dissolved O₂ from GHEA over 20 days under laser irradiation.

7. Response to comment #5 of reviewer #3: the flow cytometry results lack a control for M2

macrophages. Therefore, it is not convincing that the developed wound dressings can polarize macrophages toward the M2 phenotype.

Response: Thank you for your feedback. We agree that a control for M2 macrophages would strengthen the flow cytometry results and provide more convincing evidence for the polarization of macrophages toward the M2 phenotype. The relevant data have been presented in the revised manuscript.

Methods section:

Anti-Inflammatory Capacity

In a macrophage model induced by LPS+IFN- γ or IL4, the anti-inflammatory capacity was evaluated. During macrophage culture, 100 ng/mL LPS and 10 ng/mL IFN- γ were added to the culture medium and cultured for 12 hours. Subsequently, GHEA@Gel, RHEA@Gel, and G-RHEA@Gel were added to the culture medium and continued to grow for 24 hours. Macrophages were fixed with 4% paraformaldehyde. Then, immunofluorescence staining and FCM analysis for CD206 and CD86 were performed. During this process, macrophages coincubated with 50 ng/mL IL4 induced the M2 phenotype as a control.

Results section:

To further validate the promoting effect of AST on the transformation of M1 macrophages into M2 macrophages, flow cytometry was used to detect macrophages treated with different groups. The results shown in Figure 6d were consistent with the confocal microscopy results (Figure 6d).

Figure 6d. Flow cytometry analysis of macrophage polarization under different treatments.

8. Response to comment #6 of reviewer #3: the results of anti-*Staphylococcus aureus* antibody staining (Fig. 7h) is not convincing as no signal can be seen. This result is also inconsistent with that in Figure S19. Therefore, it is questionable whether wound dressings have antibacterial properties in vivo.

Response: Thank you for your comments, and we apologize for the confusion. To obtain more informative experimental results, we replicated the experiment and adjusted the antibody staining protocol to enable a clear observation of the signal for *Staphylococcus aureus* (as indicated by green fluorescence in the images). The newly obtained results have been included in the revised manuscript.

Results section:

In addition, immunohistochemical staining using an anti-*Staphylococcus aureus* antibody was applied to assess bacterial contamination of wounds. As shown in Figure 7h, on day 20, the control group and pure gel group wound tissue contained a high number of bacteria (green fluorescence).

Figure 7h. Immunohistochemical staining of the wound on day 20.

Reviewer #2:

The revised manuscript could be publication right now.

Response: We appreciate your feedback and will make sure to double-check everything before submitting it for consideration. Thank you for your guidance and support.

Reviewers' Comments:

Reviewer #1:

Remarks to the Author:

After re-evaluating the manuscript, I would like to provide the following comments regarding the safety, translational potential, and animal welfare considerations associated with this manuscript:

The inclusion of live *Haematococcus* in the wound dressing is acknowledged as a key contributor to its therapeutic efficacy. However, there is ambiguity regarding the method for removing *Haematococcus* after wound repair. Additionally, the long-term impact of *Haematococcus* on skin cells has not been explored in this study, leaving uncertainties about its safety potential.

The authors assert that live *Haematococcus* exhibits universal functions, including antibacterial activity, oxygen supply, ROS scavenging, and M2 macrophage polarization. Nevertheless, there is a lack of clarity on how precise control can be maintained to simultaneously harness these multifaceted functions. Consequently, the translational potential of this approach remains uncertain.

The manuscript falls short in providing detailed information on animal welfare considerations, particularly in the section related to Infected Diabetic Wound Healing. A more comprehensive description of the steps taken to ensure the welfare of the animals involved in the study is warranted.

Reviewer #1:

After re-evaluating the manuscript, I would like to provide the following comments regarding the safety, translational potential, and animal welfare considerations associated with this manuscript:

Response: We very much appreciate the reviewer's thoughtful and professional comments. We have followed the reviewer's comments and performed additional experiments to address the points raised by the reviewer. Please see our point-by-point responses below.

1. The inclusion of live *Haematococcus* in the wound dressing is acknowledged as a key contributor to its therapeutic efficacy. However, there is ambiguity regarding the method for removing *Haematococcus* after wound repair. Additionally, the long-term impact of *Haematococcus* on skin cells has not been explored in this study, leaving uncertainties about its safety potential.

Response: Thank you for your thoughtful and helpful comments. We acknowledge the ambiguity regarding the method for removing *Haematococcus* after wound repair and the lack of exploration into the long-term impact of *Haematococcus* on skin cells in our study. Before the reviewer raised this issue, we had already started long-term safety testing on live *Haematococcus*-based hydrogel. In addition, additional details about the method for removing *Haematococcus* after wound repair have been added to our revised Methods section. Thank you for bringing these important points to our attention.

Methods section:

Cell Toxicity

HaCaTs were utilized to assess the cytotoxicity of the hydrogels via an MTT assay. The cells were seeded into the lower chamber of a 24-well transwell plate (each well containing 1×10^5 cells) for 12 h. Subsequently, the G-RHEA@Gel hydrogel was added to each upper chamber and incubated for another 12 h. Then, MTT solution was added, and the cells were incubated for 2 h. The absorbance at 490 nm was measured via a standardized process. Additionally, calcein-AM (4×10^{-6} M) and PI solutions (4×10^{-6} M) were used to distinguish the distribution of living and dead cells. Cell apoptosis was analyzed via confocal laser scanning microscopy.

Long-term Impact of the G-RHEA@Gel on the Skin

The safety potential of a material is critical to its application. To explore the long-term impact of HEA@Gel on skin, the healthy skin of mice was covered with the G-RHEA@Gel hydrogel for 30 days. Then, the mice were euthanized by CO₂ inhalation followed by cervical dislocation, and the skin covered with hydrogel was collected for H&E and Masson staining. In addition, the TNF- α , IL-1 β , IL-4, IL-6, IL-10, CXCL-1, and Gran levels as well as WBC and Lymph counts were also detected.

Infected Diabetic Wound Healing

After the infected diabetic wound model was established, droplets of hydrogel solution, including hydrogel, GHEA@Gel, RHEA@Gel, or G-RHEA@Gel, were added to the wound. The fluid from the hydrogel was used to quickly cover the entire wound area, after which the hydrogel was irradiated under an ultraviolet lamp for 10 seconds to achieve hydrogel curing. The mice were kept under anesthesia while the hydrogel was covered and photocuring. The wounds were treated with PBS as a control group. After that, the wounds with different hydrogel covers were exposed to a 658 nm laser with an intensity of 0.5 W/cm^2 for 5 min to achieve sterilization through photothermal conversion. The mice were kept under anesthesia during high-power laser (0.5 W/cm^2) irradiation. Every subsequent day, the mice were anesthetized with isoflurane anesthetic, and a 658 nm laser

was used to illuminate the wound at a light intensity of 0.1 W/cm^2 for 30 min in the following 10 days. Low-intensity laser irradiation has two effects. First, GHEA activates photosynthesis to produce oxygen to improve the hypoxic wound microenvironment and accelerate vascular regeneration and wound healing. The second is to promote the release of AST from RHEA to reduce the levels of overexpressed ROS and inflammatory factors and regulate the immune microenvironment in wounds. As the wound healed, the edge of the hydrogel gradually separated from the healed wound tissue, and the hydrogel was slowly removed from the skin on the 12th day after the mice were anesthetized with isoflurane.

Results section:

Furthermore, to assess the safety profile of the hydrogel, G-RHEA@Gel was incubated with HaCaT cells. The results demonstrated excellent safety, as evidenced by the cell activity following coincubation with G-RHEA@Gel. Figure S25a shows that, compared with the control treatment, the G-RHEA@Gel treatment did not adversely affect cell activity. Additionally, staining the cells with calcein-AM and PI dyes and observing them using a confocal laser scanning microscope intuitively revealed that the cells maintained good activity after coincubation with G-RHEA@Gel (Figure S25b). Moreover, the healthy skin of the mice was covered with the G-RHEA@Gel hydrogel for 30 days. Histological examination via H&E and Masson staining revealed no evidence of skin damage following prolonged exposure to G-RHEA@Gel on the skin surface (Figure S26). To further explore the long-term safety of G-RHEA@Gel, the levels of inflammatory factors, including TNF- α , IL-1 β , IL-4, IL-6, IL-10, and CXCL-1, and hematological indices, including WBC, lymphocyte, and granule counts, in the peripheral blood of the mice were detected. As shown in Figure S27, there was no significant difference between the G-RHEA@Gel-treated mice and the control group.

Figure S25. (a) The biocompatibility of G-RHEA@Gel for HaCaTs was assessed by MTT assays. The data are presented as the mean \pm s.d. ($n = 5$ biologically independent cells). (b) Live/dead staining images of HaCaTs treated with G-RHEA@Gel. Scale bars, 200 μm . Each experiment was repeated independently three times with similar results.

Figure S26. H&E and Masson staining images of skin after different treatments. Scale bars, 500 μm . Each experiment was repeated independently three times with similar results.

Figure S27. (a) Analysis of inflammatory factors in the peripheral blood. Data are presented as the mean \pm s.d. ($n = 5$ biologically independent mice). (b) Blood hematology analysis of the mice. Data are presented as the mean \pm s.d. ($n = 5$ biologically independent mice). Statistical differences were analyzed by Student's two-sided t test.

2. The authors assert that live *Haematococcus* exhibits universal functions, including antibacterial activity, oxygen supply, ROS scavenging, and M2 macrophage polarization. Nevertheless, there is a lack of clarity on how precise control can be maintained to simultaneously harness these multifaceted functions. Consequently, the translational potential of this approach remains uncertain.

Response: Thank you for your feedback. We understand the need for precise control of live *Haematococcus* species to harness their multifaceted functions. In our revised manuscript, we have provided more detailed information on the methods and techniques used to maintain precise control over the functions of live *Haematococcus*. We believe that this additional information will address the uncertainty regarding the translational potential of this approach.

Methods section:

Infected Diabetic Wound Healing

After the infected diabetic wound model was established, droplets of hydrogel solution, including hydrogel, GHEA@Gel, RHEA@Gel, or G-RHEA@Gel, were added to the wound. The fluid from the hydrogel was used to quickly cover the entire wound area, after which the hydrogel was

irradiated under an ultraviolet lamp for 10 seconds to achieve hydrogel curing. The mice were kept under anesthesia while the hydrogel was covered and photocuring. The wounds were treated with PBS as a control group. After that, the wounds with different hydrogel covers were exposed to a 658 nm laser with an intensity of 0.5 W/cm² for 5 min to achieve sterilization through photothermal conversion. The mice were kept under anesthesia during high-power laser (0.5 W/cm²) irradiation. Every subsequent day, the mice were anesthetized with isoflurane anesthetic, and a 658 nm laser was used to illuminate the wound at a light intensity of 0.1 W/cm² for 30 min in the following 10 days. Low-intensity laser irradiation has two effects. First, GHEA activates photosynthesis to produce oxygen to improve the hypoxic wound microenvironment and accelerate vascular regeneration and wound healing. The second is to promote the release of AST from RHEA to reduce the levels of overexpressed ROS and inflammatory factors and regulate the immune microenvironment in wounds. As the wound healed, the edge of the hydrogel gradually separated from the healed wound tissue, and the hydrogel was slowly removed from the skin on the 12th day after the mice were anesthetized with isoflurane.

Results section:

In Vivo Evaluation of the Healing of Infected Diabetic Wounds

The therapeutic approach is depicted in Figure 7a. Benefiting from the photothermal conversion effect of HEA@Gel under 658 nm laser irradiation at an intensity of 0.5 W/cm² and the nonsymbiosis effect between HEA and *S. aureus*, the infected diabetic wound was sterilized first. After bactericidal treatment, every subsequent day, a 658 nm laser was used to illuminate the wound at a light intensity of 0.1 W/cm² for 30 min in the following 10 days. Low-intensity laser irradiation has two effects. First, GHEA activates photosynthesis to produce O₂ to improve the hypoxic wound microenvironment and accelerate vascular regeneration and wound healing. The second is to promote the release of AST from RHEA to reduce the levels of overexpressed ROS and inflammatory factors and regulate macrophage M2 polarization in wounds. Hence, HEA@Gel likely promoted fibroblast proliferation, keratinocyte migration, endothelial cell differentiation, and the progressive acceleration of infected wound healing in diabetic patients.

3. The manuscript falls short in providing detailed information on animal welfare considerations, particularly in the section related to Infected Diabetic Wound Healing. A more comprehensive description of the steps taken to ensure the welfare of the animals involved in the study is warranted.

Response: We appreciate your concerns regarding the welfare of the animals and have taken them into consideration. We have revised the manuscript to include a more comprehensive description of the steps taken to ensure the welfare of the animals involved in the study, particularly in the section related to Infected Diabetic Wound Healing. Thank you for bringing this to our attention.

Methods section:

Animal study

Female BALB/c mice (6 weeks, 14–16 g) provided by Beijing HFK Bioscience Company were used in this study. All animals were housed in a specific pathogen-free (SPF) animal facility for two weeks for environmental adaptation and allowed free access to food and water. During the experiment, all animals were kept in the same standard environment (23–26°C, 40–60% humidity, 12 h light–dark cycle, and 4 mice/cage). All procedures, including animal care, wound modeling, dosing, and termination, were performed according to the Experimental Animal Guidelines for Ethical Review of Animal Welfare (GB/T 35892-2018) and approved by the Tianjin University

Institute Animal Ethics Committee with the assigned approval number TJUE-2023-236. The mice were anesthetized with isoflurane anesthetic before any procedure that would cause pain. After the experiment, the mice were euthanized by CO₂ inhalation followed by cervical dislocation.

Establishment of an infected diabetic wound model

To evaluate the therapeutic efficacy of HEA@Gel, a mouse model of infected diabetic wounds was created. Mice weighing 25–30 g were fasted for 12 hours, followed by intraperitoneal injection of streptozotocin (40 mg/kg), and the process was repeated three times within three days. The mice were kept under anesthesia during intraperitoneal injection. All mice were provided with 10% sucrose water. Two weeks after the third injection, diabetic mice were identified as having blood glucose levels exceeding 16.1 mmol/L for two consecutive measurements within two days. The diabetic mice were anesthetized with isoflurane anesthetic, and their back fur was completely shaved. Then, 10 mm-diameter skin biopsy punches were used to create full-thickness wounds, and 10 μ L of *S. aureus* suspension (1×10^7 CFU/mL) was dropped on the surface of the wounds to create an infected wound model. The mice were kept under anesthesia during this procedure. After one day of infection, the number of bacterial colonies increased from 10^5 to 10^6 CFUs, indicating that the infection model was successfully established.

Infected Diabetic Wound Healing

After the infected diabetic wound model was established, droplets of hydrogel solution, including hydrogel, GHEA@Gel, RHEA@Gel, or G-RHEA@Gel, were added to the wound. The fluid from the hydrogel was used to quickly cover the entire wound area, after which the hydrogel was irradiated under an ultraviolet lamp for 10 seconds to achieve hydrogel curing. The mice were kept under anesthesia while the hydrogel was covered and photocuring. The wounds were treated with PBS as a control group. After that, the wounds with different hydrogel covers were exposed to a 658 nm laser with an intensity of 0.5 W/cm^2 for 5 min to achieve sterilization through photothermal conversion. The mice were kept under anesthesia during high-power laser (0.5 W/cm^2) irradiation. Every subsequent day, the mice were anesthetized with isoflurane anesthetic, and a 658 nm laser was used to illuminate the wound at a light intensity of 0.1 W/cm^2 for 30 min in the following 10 days. Low-intensity laser irradiation has two effects. First, GHEA activates photosynthesis to produce oxygen to improve the hypoxic wound microenvironment and accelerate vascular regeneration and wound healing. The second is to promote the release of AST from RHEA to reduce the levels of overexpressed ROS and inflammatory factors and regulate the immune microenvironment in wounds. As the wound healed, the edge of the hydrogel gradually separated from the healed wound tissue, and the hydrogel was slowly removed from the skin on the 12th day after the mice were anesthetized with isoflurane.

Reviewers' Comments:

Reviewer #1:

Remarks to the Author:

I would recommend to accept the revised work for publication.

REVIEWERS' COMMENTS:

Reviewer #1 (Remarks to the Author):

I would recommend to accept the revised work for publication.

Response: Thank you very much for your kind words after revision.